# E-BATS: Efficient Backpropagation-Free Test-Time Adaptation for Speech Foundation Models

**Jiaheng Dong**
The University of Melbourne
jiaheng.dong@student.unimelb.edu.au

**Hong Jia**
University of Auckland
hong.jia@auckland.ac.nz

**Soumyajit Chatterjee**
Nokia Bell Labs, UK
soumyajit.chatterjee@nokia-bell-labs.com

**Abhirup Ghosh**
University of Birmingham
a.ghosh.1@bham.ac.uk

**James Bailey**
The University of Melbourne
baileyj@unimelb.edu.au

**Ting Dang**
The University of Melbourne
Ting.Dang@unimelb.edu.au

## Abstract

Speech Foundation Models encounter significant performance degradation when deployed in real-world scenarios involving acoustic domain shifts, such as background noise and speaker accents. Test-time adaptation (TTA) has recently emerged as a viable strategy to address such domain shifts at inference time without requiring access to source data or labels. However, existing TTA approaches, particularly those relying on backpropagation, are memory-intensive, limiting their applicability in speech tasks and resource-constrained settings. Although backpropagation-free methods offer improved efficiency, existing ones exhibit poor accuracy. This is because they are predominantly developed for vision tasks, which fundamentally differ from speech task formulations, noise characteristics, and model architecture, posing unique transferability challenges. In this paper, we introduce E-BATS, the *first* Efficient BAckpropagation-free TTA framework designed explicitly for speech foundation models. E-BATS achieves a balance between adaptation effectiveness and memory efficiency through three key components: (i) lightweight prompt adaptation for a forward-pass-based feature alignment, (ii) a multi-scale loss to capture both global (utterance-level) and local distribution shifts (token-level) and (iii) a test-time exponential moving average mechanism for stable adaptation across utterances. Experiments conducted on four noisy speech datasets spanning sixteen acoustic conditions demonstrate consistent improvements, with $4.1\%$–$13.5\%$ accuracy gains over backpropagation-free baselines and $2.0\times$–$6.4\times$ GPU memory savings compared to backpropagation-based methods. By enabling scalable and robust adaptation under acoustic variability, this work paves the way for developing more efficient adaptation approaches for practical speech processing systems in real-world environments. Code is available at: https://github.com/JiahengDong/E-BATS

## 1 Introduction

Speech foundation models (SFM), large-scale pre-trained models that learn generalized representations from vast amounts of unlabeled speech data, have shown strong performance for a wide range of applications including voice assistants [1], transcription services [2], and accessibility tools [3]. These systems generally rely on the assumption that the training and test data follow the same distributions. In practice, this assumption is often violated, leading to significant performance drops under domain shifts caused by real-world acoustic variations such as background noise, speaker accents, and microphone characteristics [4]. While domain adaptation [5, 6, 7, 8] and domain gen-

39th Conference on Neural Information Processing Systems (NeurIPS 2025).

eralization [9, 10, 11, 12] have been extensively studied to address distributional shifts, they often require access to labeled target domain data or continuous availability of raw source data. These requirements are seldom feasible in real-world scenarios following model deployment. Recently, Test-Time Adaptation (TTA) has emerged as an attractive solution, adapting pre-trained models to new domains during inference using only unlabeled test data.

Existing TTA methods can be broadly categorized into backpropagation-based (BP) and backpropagation-free (BP-free) approaches. While the former achieved state-of-the-art (SOTA) performance using entropy minimization [13, 14, 15, 16, 17, 18, 19] or pseudo-labeling techniques [20, 21], they have a large memory overhead, mainly due to gradient computation. Even when updates are limited to a small subset of model parameters, such as batch normalization layers [13, 14, 15] or early exits [22], these methods still require high GPU memory due to automatic differentiation frameworks. This significantly limits their practical use in continuous inference scenarios and on resource-constrained devices. In contrast, BP-free TTA methods eliminate the need for gradient computation, making them more efficient and computationally lightweight. These methods either modify the model parameters during the forward pass [23, 24, 25, 26, 27] or learn a new input prompt, a vector integrated with the partially processed input samples at an intermediate layer of the model [28].

Despite the promise of TTA, they are largely tailored to models that depend on Batch Normalization (BN), while SFMs use Layer Normalization (LN), limiting the applicability of BN-based TTA [16]. Additionally, SFMs include both CNN-based feature encoders extracting localized spectral features and transformer encoders processing global context. This is unlike models in other modalities like vision, which are either CNN-based (e.g., ResNet [29]) or transformer-based (Vision Transformer [30]). Such architectural difference presents a fundamental challenge for BP-free feature adaptation. Furthermore, downstream tasks and noise characteristics differ significantly between vision and speech tasks. As shown in Figure 1, vision models are commonly used for image classification, a one-to-one mapping task where noise typically appears as spatial perturbations of pixels [31].

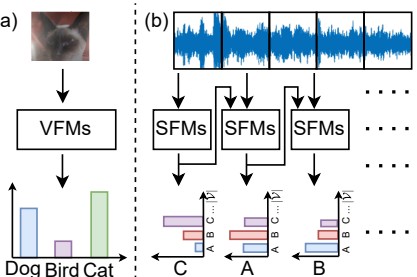

Figure 1: The main difference between (a) Vision Foundation Models and (b) Speech Foundation Models (SFMs) is the sequential pipeline in SFMs that processes a fixed-length frame of an utterance as an input and maps to a distribution over $|\mathcal{V}|$ token classes.

In contrast, speech recognition involves sequence-to-sequence mapping and must handle dynamic, temporally varying noise across frames [32]. This requires more dynamic, multi-scale adaptation. Lastly, TTA methods often depend on large batch sizes for reliable adaptation, whereas TTA in speech tasks needs to process one utterance at a time (batch size of 1) due to the high variability across speech utterances [16, 18]. Despite recent developments in TTA for SFMs [16, 19, 18, 17, 21], they still heavily rely on the BP-based TTA methods with high computational overhead, and overlook the unique requirements of multi-scale adaptation.

To address these challenges, we propose *the first Efficient BAckpropagation-free single-utterance TTA framework for SFMs*, E-BATS, which achieves SOTA accuracy with memory efficiency. Here we focus on one of the most popular tasks on speech data – speech recognition. E-BATS consists of three novel modules: i) A lightweight, prompt-based tuning mechanism tailored for SFMs which directly adapts latent feature distribution using forward pass only; ii) a custom multi-scale loss function that captures both global (utterance-level) and local (token-level) latent embeddings distribution shifts; and iii) a test-time Exponential Moving Average (T-EMA) module that stabilizes prompt updates across dynamic utterances. The main contributions are threefold:

- We introduce the first backpropagation-free TTA approach tailored explicitly for SFMs that achieves high accuracy and low memory consumption.
- We propose a novel framework E-BATS, consisting of three novel modules to effectively address the multi-scale domain shifts and stable adaptation across the dynamic speech streams.
- We validate E-BATS across four noisy datasets, sixteen acoustic environments, and two model architectures have demonstrated significant improvements in both accuracy and memory, particularly $2\times$ to $6.4\times$ reduction in peak GPU memory usage over SOTA baselines.

## 2 Related Works

**Memory-Intensive BP TTA Methods.** Traditional backpropagation-based TTA methods are generally memory-intensive, as they rely on gradient-based updates of model parameters, typically guided by entropy minimization or pseudo-labeling. TENT [13] was the first to adapt the affine parameters of the BN layers using entropy minimization. SAR [15] and EATA [14] are variants of TENT, which further filter out a small portion of data samples that are unreliable and redundant. Although they only update a small portion of the overall parameters, the computation is still overhead due to backpropagation and the large batch size required for reliable adaptation [15]. More advanced methods, like CoTTA [20], employ additional networks (e.g., teacher-student models) for adaptation, but at the cost of significantly increased computational and memory overhead.

**TTA for SFMs.** A few recent TTA approaches have been tailored for SFMs, which typically extend memory-intensive BP TTA methods with speech-specific mechanisms [16, 18, 19]. SUTA [16] built upon TENT by updating CNN-based feature encoder layers alongside normalization parameters, incorporating techniques like temperature smoothing and Minimum Class Confusion loss. SGEM [18] and CEA [19] further improved upon this with advanced loss design for audio tasks, such as sequence-level entropy minimization or uncertainty-driven frame prioritization. DSUTA [17] and AWMC [21] introduced additional subnetworks (e.g., fast-slow models and anchor-chaser-leader models) to enhance cross-utterance knowledge transfer. However, these methods still heavily rely on backpropagation, leading to significant memory overhead and scalability challenges as more layers are updated beyond normalization.

**BP-free TTA Methods.** BP-free TTA methods offer an efficient alternative by updating the model solely via forward pass to achieve computational efficiency. These methods generally fall into three categories: (i) analytical adjustment of batch normalization statistics [23, 24, 25, 26], (ii) adaptation of the classifier using class prototypes [27] or output probabilities [33], and (iii) optimization via evolutionary algorithms that circumvent gradient-based updates [28]. However, they typically offer lower adaptation accuracy compared to memory-intensive BP TTA methods [13, 20, 14, 15]. Currently, *BP-free TTA for speech foundation models remains unexplored*, primarily due to the unique challenges posed by sequence-to-sequence learning, single-utterance adaptation rather than batches, and differences in model structures, particularly in normalization layers and feature encoders.

## 3 Methodology

### 3.1 Overview

We consider a *covariate shift* between the source and a target domain, such that the marginal distributions of speech differ, $P_{\text{src}}(\boldsymbol{X}) \neq P_{\text{tgt}}(\boldsymbol{X})$, while the class prior $P_{\text{src}}(y) = P_{\text{tgt}}(y)$ and the conditional distribution $P_{\text{src}}(y \mid \boldsymbol{X}) = P_{\text{tgt}}(y \mid \boldsymbol{X})$ are preserved. We adapt an SFM $\Theta_{\text{src}}$, pre-trained on a source speech dataset $D_{\text{src}}$, to improve transcription accuracy while maintaining the low peak memory usage on an unlabeled target speech dataset $D_{\text{tgt}}$. A target stream $D_{\text{tgt}} = \{\boldsymbol{X}_1, \ldots, \boldsymbol{X}_T\}$ consists of $T$ utterances arriving sequentially, each is processed under an online, single-utterance setting with a batch size of one. Each utterance $\boldsymbol{X}_t$ is composed of a variable number of frames $N_t$ and is represented as a sequence of frame-level feature vectors, where each frame represents a short, fixed-duration segment of the audio signal: $\boldsymbol{X}_t = \left[ \boldsymbol{x}_t^1, \boldsymbol{x}_t^2, \ldots, \boldsymbol{x}_t^{N_t} \right]$, $\boldsymbol{x}_t^i \in \mathbb{R}^{d_{\text{in}}}$. The model $\Theta_{\text{src}}$ is composed of two components, $\Theta = g \circ h$, where: (a) $h$ is a convolutional encoder that maps each input frame to a latent embedding: $\boldsymbol{x}_t^i \mapsto \boldsymbol{z}_t^i \in \mathbb{R}^d$. (b) $g$ consists of a stack of transformer layers and a Connectionist Temporal Classification (CTC) classifier head, producing frame-level posterior distributions over CTC token classes $v \in \mathcal{V}$, where $\mathcal{V}$ consists of twenty-six alphabet token classes (a–z) and six special token classes (apostrophe, blank, etc.): $g(\boldsymbol{Z}_t) = \left\{ P(y_t^i \mid \boldsymbol{z}_t^{1:N_t}) \right\}_{i=1}^{N_t}$. These posteriors are further decoded to produce each utterance's final transcription $\hat{y}_t$.

**System Overview.** As shown in Figure 2, for a test-time utterance $\boldsymbol{X}_t$, the model first employs a Lightweight Prompt Adaptation (LPA) module, which directly modifies the CNN encoder features $\boldsymbol{Z}_t$ by incorporating $J$ learnable prompt vectors $\boldsymbol{s}_{t,j}$. These prompts are sampled using the derivative-free Covariance Matrix Adaptation Evolution Strategy (CMA-ES), guided by a multi-scale adaptation loss $L_{\text{adapt}}$. The prompt that results in the lowest loss is selected for adaptation. The loss comprises three components: entropy minimization ($L_{\text{ent}}$), utterance-level ($L_{\text{utt}}$), and token-level

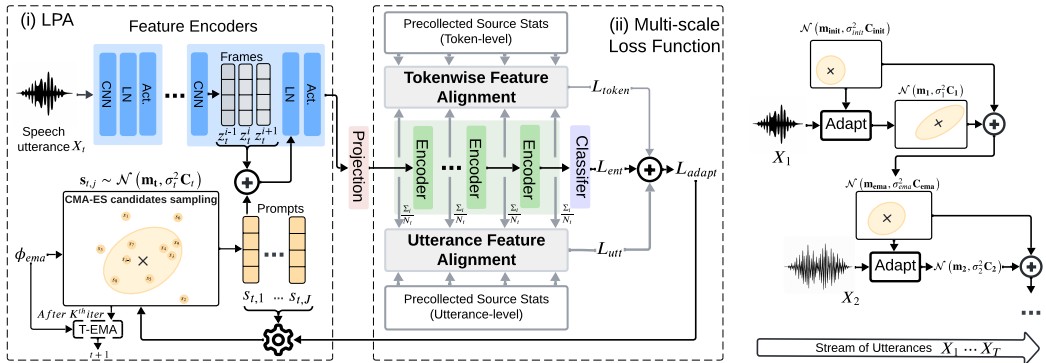

(a) Single-utterance Backpropagation-free Adaptation      (b) T-EMA across utterances

Figure 2: Overall framework of E-BATS. For an utterance $X_t$: (i) *Lightweight Prompt Adaptation (LPA)*: CNN-extracted latent features $Z_t$ are adapted using a set of $J$ candidate prompts $s_{t,j}$ generated by CMA-ES in parallel, leading to $J$ adapted representations. (ii) The adapted representations $\mathbf{J}$ are evaluated, and their corresponding prompts are ranked using a multi-scale loss (entropy loss, utterance-level and token-level feature alignment). This ranking guides the iterative update of CMA-ES parameters over $\mathbf{K}$ iterations until the loss converges, at which point the best prompt is selected for adaptation. The CMA-ES parameters are smoothed using T-EMA for next utterance adaption. (b) *Test-time Exponential Moving Average (T-EMA)*: T-EMA stabilizes adaptation by smoothing the CMA-ES search trajectory across a stream of utterances, facilitating robust prompt learning.

latent embeddings alignment ($L_{\text{token}}$). To promote stable adaptation across consecutive utterances, a Test-time Exponential Moving Average (T-EMA) module (Figure 2(b)) incrementally updates the CMA-ES search distribution, enabling smoother evolution of the prompt vector $s_{t,j}$ over time.

## 3.2 Lightweight Prompt Adaptation (LPA)

To ensure memory efficiency, E-BATS adopts prompt tuning [34]. This technique introduces a small set of learnable parameters, called *prompts*, to guide the behavior of a pre-trained model on a downstream task while keeping the original model weights fixed. While conventional prompt tuning approaches have primarily been developed for transformer-only architectures [30], typically by concatenating prompts with the model's inputs, we propose a novel LPA module (Figure 2a) designed specifically for SFMs that include convolutional components. The LPA integrates adaptive prompts directly into the convolutional layers, leveraging their effectiveness in extracting acoustic features [16]. Furthermore, rather than common strategy of concatenating prompts with input features [28], our approach examines latent feature distribution shifts and leverages shifting patterns to guide adaptation.

**Characterizing Distribution Shifts.** To understand differences in speech distribution, we first characterize the shift between the embedding vectors produced by source $P(Z_{\text{src}})$ and a target domain $P(Z_{\text{tgt}})$. We study the two most natural ways, covering both first and second order statistics: $a)$ comparing shift in the centroid of the point clouds, and $b)$ comparing the spread of the point clouds using their covariances. Incidentally, the popular distance measure between distributions, Fréchet Inception Distance, is a linear combination of these two factors, justifying our choice. Figure 3 shows that, under various real-world noise and variability conditions, the shift in the mean accounts for up to

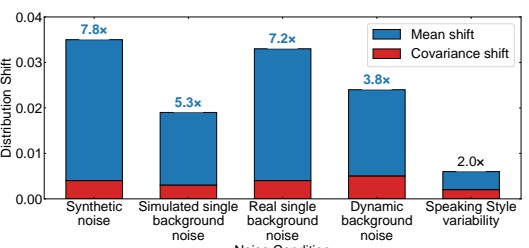

Figure 3: Comparing the source and target latent spaces across different acoustic conditions (same sample size for source and target domain within each condition). Blue and red bars indicate the mean and covariance shifts.

$7.8\times$ of the shift in the covariance (we make sure they are in comparable scale, explained in Appendix A).

From this observation, we hypothesize that the shift between $D_{src}$ and $D_{tgt}$ can be explained by a simple geometric translation operation in the latent space; therefore, an adequate shift of target embeddings, $Z_t$ would align the latent target vectors to the ones from the source, essentially mitigating the problem of domain shift. It is important to note that, despite the potential for a translation operation, this remains a non-trivial problem. In real-world scenarios, noise is not consistently introduced into clean samples, making it impossible to derive a simple solution based on an analyzable shifting vector. Therefore, adaptively learning the prompt becomes necessary.

**Learning Prompt to Adapt.**    Following the above observation, we propose to add the embeddings $Z_t$ and the prompt vector $s_t$ over all frames (Figure 2a) as:

$$\hat{Z}_t = Z_t + s_t \cdot \mathbf{1}_N^\top, \tag{1}$$

where $Z_t = \left[ z_t^1, \ldots, z_t^{N_t} \right]$, $s_t \in \mathbb{R}^d$ is the prompt vector for $X_t$. We exclusively optimize the prompt vector for each utterance while the rest of the model parameters remain frozen.

**Prompt Optimization with CMA-ES.**    To identify the optimal prompt vector in a backpropagation-free manner, we adopt the CMA-ES [35]. For each test-time utterance $X_t$, CMA-ES samples $J$ candidate prompt vectors $s_{t,j}$ from a multivariate normal distribution $\mathcal{N}(.)$, which are parameterized by a mean vector $\mathbf{m}_t \in \mathbb{R}^d$, a covariance matrix $\mathbf{C}_t \in \mathbb{R}^{d \times d}$, and a step size $\sigma_t \in \mathbb{R}_{>0}$. The performances of these candidates are then ranked based on a loss function (explained in Section 3.3), which informs the update of the distribution parameters $m_t^{(k)}, C_t^{(k)}, \sigma_t^{(k)}$ for the next iteration $k$. The step size scales the covariance and thus controls the spread of candidate samples drawn from the distribution in the next iteration. The iteration continues until the loss convergences. The final prompt vector $s_t$ is selected as the candidate that minimizes the loss for adaptation (Appendix D):

$$s_t = \arg \min_{s_{t,j} \in \mathbb{R}^d} L_{adapt}(X_t, s_{t,j}). \tag{2}$$

### 3.3   Multi-scale Loss Function

The loss function $L_{adapt}$ that effectively guides the prompt optimization is proposed to integrate entropy minimization ($L_{ent}$) with feature alignment to the source domain latent embedding distributions at multiple scales, i.e., utterance-level ($L_{utt}$) and tokenwise ($L_{token}$). We optimize a weighted average using $L_{adapt} = \alpha L_{ent} + \beta L_{utt} + c L_{token}$ where $\alpha$ and $\beta$ are hyperparameters, and $c$ is chosen algorithmically based on the confidence in the prediction.

**Entropy Minimization Loss with Blank Token Exclusion ($L_{ent}$).**    Entropy minimization aims to improve prediction confidence, which is further improved with blank token exclusion for SFMs. SFMs for speech recognition include a special 'blank' token class $\varnothing$, which is primarily designed in CTC to address the alignment mismatch between input frames and output labels [36] with different lengths. In recent studies, it often indicates frames where no alphabetic character can be assigned, helping to eliminate the need to identify ambiguous sound boundaries [36] and highlighting silent periods [37]. In practice, predictions for a large proportion of the frames often belong to this blank class, which introduces class imbalance [16]. We address this by defining the entropy loss only considering the set of frames $\tilde{X}_t$ that do not predict blanks. Formally, we use Shanon's entropy as $L_{ent} = -\frac{1}{|\tilde{X}_t|} \sum_{x_t^i \in \tilde{X}_t} \mathcal{H}(\Theta(x_t^i))$.

Despite its effectiveness, minimizing only $L_{ent}$ has a trivial solution of predicting the blank token class for each frame (as highlighted in Section 5.5). Thus, we introduce an utterance-level latent embeddings alignment loss to guide optimization towards the source domain embeddings correctly.

**Utterance-level Latent Embeddings Alignment ($L_{utt}$).**    Utterance-level latent embeddings alignment aims to align the *global latent embedding distributions* between the source domain and the target domain to avoid trivial solutions. At each transformer encoder layer, $l$, we compute the squared Euclidean distance between the source-domain centroid and the target-domain centroid of utterance-level latent embeddings (Figure 2a(ii)). Each utterance-level embedding is obtained by averaging the embeddings across all frames within that utterance. Effectively, we compute and store the centroids, $\boldsymbol{\mu}_{src}$ using $D_{src}$ and $\Theta_{src}$ and compare against the centroids, $\boldsymbol{\mu}_{tgt}$, computed from the current target utterance. This gives us the following loss component: $L_{utt} = \frac{1}{L} \sum_{\ell=0}^{L} \|\boldsymbol{\mu}_{tgt}^l - \boldsymbol{\mu}_{src}^l\|_2^2$, where $L$ is the number of transformer layers in $\Theta$. Note that $\boldsymbol{\mu}_{src}$ requires a small storage size, in the order of $L \times d$, and as this does not correspond to an individual source sample, it is privacy preserving.

**Adaptive Confidence Tokenwise Latent Embeddings Alignment ($L_{token}$).** The tokens within a single utterance may not comprehensively represent all possible tokens. Aligning the centroids of the source and target utterance-level embeddings could lead to bias towards the majority tokens. To address this, we minimize the distance between the source and target latent distributions corresponding to token classes, where target token classes are estimated using pseudo-labels. To prevent unreliable pseudo-labels from distorting adaptation, we introduce adaptive confidence coefficients for the token-level loss. Specifically, when overall distribution shifts are substantial or entropy is high, this indicates less reliable posterior probabilities and a higher risk of inaccurate token pseudo-labels. In such cases, the confidence for token-level loss is reduced to minimize the impact of misleading token predictions. Conversely, if the shifts are minor or entropy is low, stronger alignment of token-level distributions can be applied with greater confidence.

Formally, we represent the mean $\boldsymbol{\mu}^{v,l}$ and the standard deviation $\boldsymbol{\sigma}^{v,l}$ of the distribution for each token class $v \in \mathcal{V}$ in a $d$-dimensional space at transformer layer $l$. Then we define the loss as the average distance between distributions as:

$$L_{token} = \frac{1}{L}\frac{1}{|\mathcal{V}|}\sum_{l=0}^{L}\sum_{v\in\mathcal{V}}\left(\left\|\boldsymbol{\mu}_{tgt}^{v,l} - \boldsymbol{\mu}_{src}^{v,l}\right\|_2^2 + \left\|\boldsymbol{\sigma}_{tgt}^{v,l} - \boldsymbol{\sigma}_{src}^{v,l}\right\|_2^2\right). \tag{3}$$

Note that the storage cost for $\boldsymbol{\mu}_{src}^{v,l}$ and $\boldsymbol{\sigma}_{src}^{v,l}$ is small and in the asymptotic order of $2 \times L \times 32 \times d$. This is privacy-preserving as the distribution parameters are sample-agnostic.

We further introduce an adaptive confidence $c \in [0, 1]$ to adjust the trustworthiness of token-level predictions. We propose using the inverse of the combined loss $H = L_{ent} + L_{utt}$ as the confidence-based coefficient, where high $H$ means lower confidence. We define the normalized coefficient using min-max normalization with predefined bounds $H_{\min}, H_{\max}$ and $c_{max}$ as: $c = c_{max} - \frac{H - H_{\min}}{H_{\max} - H_{\min} + \epsilon}$, where $\epsilon$ is a small constant to prevent division by zero. This adaptive confidence-aware scaling strengthens token-wise control via prediction reliability and domain shift.

**Optimizing through CMA-ES.** The CMA-ES parameters are optimized using $L_{adapt}$, driving the prompt optimization. By iteratively minimizing $L_{adapt}$ for candidate prompt vectors within each utterance, CMA-ES could be updated to effectively generate prompts that robustly mitigate both global (utterance-level) and local (token-level) acoustic domain shifts.

### 3.4 T-EMA across Utterances

To stabilize adaptation across the utterance streams, we propose T-EMA that updates the CMA-ES parameters incrementally, ensuring a smoother and more consistent search space for the prompt (Figure 2b). It leverages the knowledge from past utterances to initialize each new search more robustly, thereby reducing overfitting and mitigating model drift. This serves as the first smoother adaptation strategy for BP-free TTA in SFMs.

CMA-ES statistics parameters are carried over between utterances using the following weighted average scheme. We introduce EMA statistics, e.g., $\mathbf{m}_{ema}$, that are updated using a hyperparameter $\gamma \in [0, 1)$ to weight the past and current statistics values. Such an update happens when all $K$ iterations have finished for an utterance. For example, at the end of processing $t$-th utterance, the mean of the distribution is updated as $\mathbf{m}_{ema} = \gamma \mathbf{m}_{ema} + (1 - \gamma) \mathbf{m}_t^{(K)}$, where the $\mathbf{m}_{ema}$ is initialized with $\mathbf{m}_0$ when $t = 0$. Then, $\mathbf{m}_{t+1}$ is set to $\mathbf{m}_{ema}$ from the previous round for next utterance prompt learning. The other statistics, covariance and step size are updated in the same way.

## 4 Experiments

**Datasets.** We evaluate the proposed method on four datasets across sixteen acoustic conditions to assess its effectiveness under varying domain shifts. The test sets encompass three categories of acoustic variability: synthetic noise, single-domain distributional shifts, and multi-domain distributional shifts, reflecting the range of conditions encountered in real-world deployment. Following [16, 19], we introduce *synthetic noise* to the LibriSpeech `test-other` split [38] with additive Gaussian noise with zero mean and varying standard deviations ($\sigma \in \{0.0, 0.005, 0.01, 0.015, 0.02\}$) to simulate covariate shifts. We use the CHiME-3 dataset [39] representing single domain shift in a sample, including four acoustic environments: bus, café, pedestrian area, and street junction. We further use *CHiME-3-Mix* that creates a dynamic stream by concatenating CHiME-3 environments to emulate

Table 1: Word Error Rate (WER) on various noisy conditions using Wav2Vec2ForCTC-Base. Lower value means better adaptation performance. **Bold** represents the best performance for BP-free TTA, while underlined means the best for both BP-based and BP-free TTA.

| Method | BP-free | Gaussian noise | | | | | | CHiME3 (Single) | CHiME3 (Mixed) | TED | Common Voice |
|--------|---------|-----|-------|------|-------|------|------|---------|---------|-----|-------|
| | | 0.0 | 0.005 | 0.01 | 0.015 | 0.02 | Avg | | | | |
| Source | — | 8.6 | 13.9 | 24.4 | 39.5 | 54.5 | 28.2 | 34.2 | 34.2 | 13.2 | 36.8 |
| TENT | ✗ | 8.5 | 14.0 | 24.1 | 39.2 | 54.3 | 28.0 | 34.1 | 34.1 | 13.1 | 36.8 |
| EATA | ✗ | 14.1 | 18.1 | 27.0 | 37.9 | 51.3 | 29.7 | 33.1 | 39.9 | 14.1 | 61.3 |
| SAR | ✗ | 8.4 | 13.6 | 22.9 | 36.0 | 49.9 | 26.2 | 33.6 | 34.7 | 13.0 | 38.2 |
| CoTTA | ✗ | 9.2 | 12.6 | 18.1 | 39.3 | 54.5 | 26.7 | 32.9 | 34.3 | 12.8 | 36.9 |
| CEA | ✗ | 7.5 | 11.1 | 16.4 | 23.8 | 33.6 | 18.5 | 26.8 | 26.8 | 12.0 | 31.5 |
| SGEM | ✗ | 7.3 | 10.9 | 16.4 | 23.8 | 33.9 | 18.5 | 27.2 | 27.1 | 11.9 | 31.2 |
| AWMC | ✗ | 9.5 | 11.7 | 16.6 | 23.9 | 31.8 | 18.7 | 34.0 | 33.9 | 13.6 | 37.9 |
| SUTA | ✗ | 7.3 | 10.9 | 16.5 | 24.1 | 34.1 | 18.6 | 26.8 | 26.8 | 11.9 | 31.5 |
| CSUTA | ✗ | 13.1 | 17.5 | 24.5 | 31.4 | 37.0 | 24.7 | 26.5 | 32.6 | 15.6 | 135.0 |
| DSUTA | ✗ | 9.0 | 11.7 | 16.1 | 21.1 | 24.1 | 16.4 | 24.0 | 24.1 | 12.7 | 36.1 |
| T3A | ✓ | 10.0 | 15.9 | 26.8 | 42.7 | 58.6 | 30.8 | 35.9 | 35.8 | 14.6 | 38.8 |
| LAME | ✓ | 9.1 | 15.0 | 26.0 | 42.4 | 58.2 | 30.1 | 36.0 | 36.0 | 14.0 | 38.8 |
| FOA | ✓ | 8.7 | 13.9 | 22.7 | 33.3 | 45.3 | 24.8 | 31.7 | 31.1 | 13.3 | 38.2 |
| Ours | ✓ | **7.7** | **10.5** | **14.8** | **19.9** | **25.3** | **15.6** | **24.0** | **24.3** | **12.5** | **30.6** |

non-stationary acoustic shifts [17]. *CommonVoice (CV)* [40] introduces variability in speaker accents, recording devices, and environments and *TEDLIUM-v2 (TED)* [41] comprises oratory speech from TED talks with diverse accents, speaking styles, and syntactic structures.

**Baseline Methods.** We compare E-BATS against 13 SOTA TTA baselines. The **BP methods** include general approaches of Episodic TENT [13], SAR [15], EATA [14], and CoTTA [20], as well as speech-specific methods: SUTA [16], CEA [19], SGEM [18], DSUTA [17], CSUTA [17], and AWMC [21]. The **BP-free methods** include LAME [33], T3A [27], and FOA [28]. Dataset and baseline details are provided in Appendix B.

**Implementation Details.** All TTA baselines are configured for per-utterance adaptation with batch size of 1. For E-BATS, we set the CMA-ES population size $J = 50$. The loss function coefficients are $\alpha = 1.0$ and $\beta = 2.0$. We use $H_{min} = 0.0$, $H_{max} = 5.0$ in calculating the confidence-weighted coefficient $c$ with $c_{max} = 2.0$ optimized over $\{1.0, 1.5, 2.0, 2.5, 3.0, 3.5, 4.0\}$. Evaluation is performed using two commonly used SFMs, Wav2Vec2ForCTC-Base [42] and HuBERTForCTC-Large [43]; both models are fine-tuned on LibriSpeech and then are adapted in our experiments. The pre-collected statistics are sourced from clean LibriSpeech data samples. For T-EMA, we select $\gamma = 0.9$ for Wav2Vec2 and $\gamma = 0.8$ for HuBERT after tuning over $\{0.7, 0.8, 0.9, 0.95, 0.99\}$. We use Word Error Rate (WER) [44] as the evaluation metric, which measures the fraction of incorrectly predicted words in the dataset. A lower WER indicates better performance. We further conducted sensitivity analyses on key hyperparameters in Appendix C.3, including the CMA-ES population size ($J$), the number of iteration steps ($N$), the loss component weights ($\alpha, \beta$), and the T-EMA decay factor ($\gamma$), confirming that the chosen settings yield stable and robust performance across diverse configurations. All experiments are conducted on a single NVIDIA A100 GPU. Implementation details are provided in Appendix B.

## 5 Results and Discussion

### 5.1 Comparing accuracy to SOTA

Experiments using Wav2Vec2ForCTC-Base (Table 1) show that E-BATS consistently outperforms all BP-free TTA baselines across datasets, with WER reductions ranging from 0.8% to 20.0% over the strongest alternative. Its performance gains increase with noise severity, achieving at least 3.4% improvement at $\sigma = 0.005$ and 20.0% at $\sigma = 0.02$, highlighting robust adaptation under challenging conditions. T3A and LAME degrade the source model, indicating that updating only the final classifier is insufficient. FOA, which also uses prompt tuning, performs better but remains less effective than E-BATS, likely due to difficulties in adapting transformer layers for acoustic shifts (Section 5.5).

Table 2: WER on various noisy conditions using HuBERTForCTC-Large. Lower is better. **Bold**: best among BP-free; underlined: best overall.

| Method | BP-free | Gaussian noise | | | | | | CHiME3 (Single) | CHiME3 (Mixed) | TED | Common Voice |
|---|---|---|---|---|---|---|---|---|---|---|---|
| | | 0.0 | 0.005 | 0.01 | 0.015 | 0.02 | Avg | | | | |
| Source | — | 4.2 | 5.0 | 6.4 | 9.0 | 12.8 | 7.5 | 16.5 | 16.5 | 9.1 | 21.4 |
| TENT | ✗ | 4.2 | 4.9 | 6.3 | 8.8 | 12.5 | 7.3 | 16.4 | 16.4 | 9.0 | 27.5 |
| EATA | ✗ | 7.4 | 8.4 | 9.5 | 11.3 | 13.7 | 10.1 | 16.2 | 18.9 | 9.7 | 34.4 |
| SAR | ✗ | 4.0 | 4.7 | 6.3 | 8.6 | 12.2 | 7.2 | 16.4 | 17.3 | 9.0 | 21.7 |
| CoTTA | ✗ | 4.4 | 5.1 | 6.3 | 8.3 | 11.0 | 7.0 | 16.2 | 15.3 | 9.0 | 25.8 |
| CEA | ✗ | 3.8 | 4.2 | 5.1 | 6.7 | 9.1 | 5.8 | 14.2 | 14.1 | 8.1 | 18.3 |
| SGEM | ✗ | 3.7 | 5.3 | 5.3 | 6.9 | 9.3 | 6.1 | 14.2 | 14.2 | 8.3 | 18.4 |
| AWMC | ✗ | 5.5 | 6.4 | 8.2 | 10.7 | 14.3 | 9.0 | 15.9 | 17.2 | 9.9 | 21.9 |
| SUTA | ✗ | 3.8 | 4.2 | 5.1 | 6.8 | 9.2 | 5.8 | 14.2 | 14.2 | 8.2 | 18.4 |
| CSUTA | ✗ | 6.0 | 6.8 | 7.9 | 9.5 | 11.9 | 8.4 | 14.7 | 16.2 | 10.2 | 90.0 |
| DSUTA | ✗ | 4.6 | 5.0 | 6.0 | 7.1 | 8.8 | 6.3 | 13.3 | 13.5 | 8.7 | 27.4 |
| T3A | ✓ | 14.4 | 15.8 | 18.9 | 24.2 | 30.2 | 20.7 | 27.9 | 32.3 | 22.5 | 46.2 |
| LAME | ✓ | 4.5 | 5.3 | 6.9 | 9.8 | 13.9 | 8.1 | 17.5 | 17.5 | 9.7 | 22.6 |
| FOA | ✓ | 4.5 | 5.3 | 6.8 | 9.2 | 12.9 | 7.7 | 16.4 | 16.7 | **9.3** | 22.8 |
| Ours | ✓ | **4.3** | **4.9** | **5.9** | **7.5** | **9.5** | **6.4** | **14.0** | **14.0** | **9.3** | **20.1** |

Compared to BP-based methods, E-BATS achieves the lowest WER in 3 out of 5 datasets, with up to 30.7% relative improvement, and remains competitive. Methods such as EATA and CoTTA, which depend on larger batch sizes or vision-specific strategies, perform poorly across the board. While TENT and SAR are more resilient with small batches, they still underperform relative to E-BATS, showing that adapting only normalization layers is inadequate. On the other hand, the performance limitations of SGEM, SUTA, and CEA stem from their utterance-level reset strategy. This prevents them from transferring the already learned knowledge for adapting further utterances as model weights are reinitialized every time. Notably, DSUTA, despite continuous adaptation, performs 5.5% worse than E-BATS on CommonVoice, the most diverse test condition. This suggests that frequent parameter updates may lead to catastrophic forgetting. In contrast, E-BATS updates only prompt vectors, preserving the pre-trained model and enabling effective, stable adaptation across varied domains. Additional results across different fine-grained noise conditions are in Appendix C.

## 5.2 Memory Efficiency

Beyond accuracy, E-BATS demonstrates substantial memory efficiency across all evaluated methods, as shown in Figure 4. Compared to BP-based TTA methods, it reduces peak GPU memory usage by $1.5\times$ to $5.9\times$. Specifically, relative to DSUTA, CEA, and SGEM, E-BATS achieves memory savings of $3.3\times$, $3.2\times$, and $2.8\times$, respectively, while outperforming them in WER. This efficiency is attributed to lightweight prompt tuning and the T-EMA mechanism, which avoids gradient-based updates. Compared to BP-free baselines, E-BATS maintains comparable or lower memory usage while achieving a lower WER, indicating a favorable balance between efficiency and performance.

## 5.3 Performance with Different Backbone Models

When using HuBERTForCTC-Large backbone, a larger SFM than Wav2Vec2ForCTC-Base, E-BATS continues to outperform all BP-free and most BP-based TTA methods across datasets, as summarized in Table 2. On average, it achieves $1.8\%$ to $17.1\%$ lower WER than BP-free baselines. While performance is comparable to BP-based methods under certain conditions, E-BATS surpasses them in challenging scenarios such as CHiME-3 single-domain and mixed-domain settings. More notably, E-BATS offers substantially better memory efficiency at larger model scale, with $2.4\times$ to $6.8\times$ lower GPU memory usage compared to BP-based approaches (detailed in Appendix C and Figure 6). As model size increases, memory usage grows for all TTA methods; however, E-BATS scales more gracefully, exhibit-

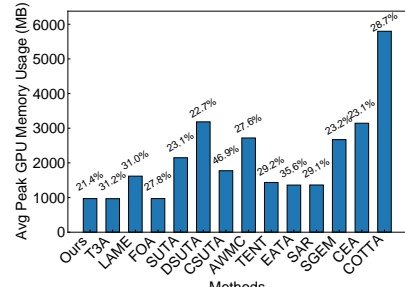

Figure 4: Balance between Average Peak GPU memory usage (bar) and average WER (percentages %) for different TTA methods across all datasets.

Table 3: Ablation study on three key components.

| Prompt Adaptation | | | Loss Function (w/ T-EMA) | | | | Loss Function (w/o T-EMA) | | | | T-EMA Mechanism | | |
|---|---|---|---|---|---|---|---|---|---|---|---|---|---|
| Feat | Trans | WER | $L_{ent}$ | $L_{utt}$ | $L_{token}$ | WER | $L_{ent}$ | $L_{utt}$ | $L_{token}$ | WER | T-EMA | Reset | WER |
| ✓ | — | 24.0 | ✓ | ✓ | ✓ | 24.0 | ✓ | ✓ | ✓ | 25.4 | ✓ | — | 24.3 |
| — | ✓ | 34.2 | ✓ | ✓ | — | 24.3 | ✓ | ✓ | — | 25.5 | — | ✓ | 26.5 |
| — | — | — | ✓ | — | — | 24.5 | ✓ | — | — | 49.6 | — | — | 25.4 |

ing only a moderate increase in memory demand. This makes it particularly suitable for on-device or resource-constrained environments where backpropagation is infeasible.

## 5.4 Memory Efficiency Across Utterance Lengths

Figure 5 shows the peak GPU memory usage as utterance duration increases on the TED dataset using the HuBERTForCTC-Large model. TED is selected due to its wide range of utterance lengths. We compare E-BATS against four top-performing BP-based TTA methods. These baselines exhibit rapidly growing memory consumption with increasing utterance duration, reaching 6–12 GB for 30-second clips. In contrast, E-BATS displays a near-linear memory profile, increasing from $\sim 1.1$ GB at 1 second to just over 1.9 GB at 35 seconds, particularly suitable for deployment scenarios with strict or varying memory constraints.

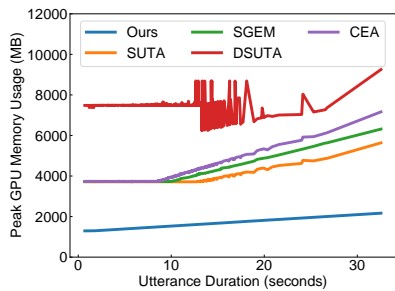

Figure 5: Peak GPU memory of TTA on TED as audio duration increases.

## 5.5 Ablation Study

We investigate the effectiveness of three key components in E-BATS.

**Prompt Adaptation.** We compare our proposed method, which injects prompts into $\mathbf{Z}_t$ to adapt the CNN latent feature representations directly, with a variant that concatenates prompts with $\mathbf{Z}_t$ at the input to the transformer encoder, following conventional prompt-tuning approaches. As shown in Table 3, adapting within the CNN-based feature encoder yields better performance (24.0 *vs.* 34.2 WER) on CHiME3 (single). This advantage comes from CNNs' ability to capture localized spectral features (e.g., pitch, formants), which are crucial for handling acoustic domain shifts. In contrast, transformer encoders focus on global contextual dependencies (e.g., sentence-level semantics), making them less effective at modeling fine-grained acoustic variations under domain mismatch.

**Loss Function Components.** We evaluated each component of the loss function under two settings: (1) CHiME-3-Single with (w/) T-EMA, representing a stable and consistent distribution shift, and (2) CHiME-3-Mix without (w/o) T-EMA, reflecting more diverse shifts. For the single-domain shift with T-EMA, we observed that each loss component contributed to overall performance (24.0 WER), with the token-level loss providing additional improvements (from 24.3). In contrast, for the mixed distributional shifts, adaptation relied heavily on utterance-level alignment (from 49.6 to 25.5), as expected due to the increased shifts. Compared with only using $L_{ent}$, combining $L_{utt}$ effectively prevents trivial solutions caused by entropy minimization of predicting all frames to the blank token class or collapsing into a single character. This is further explained and analyzed in Appendix C.4. Moreover, the adaptive weighting mechanism of the token-level loss ensured its reduced confidence, facilitating more reliable adaptation under this setting (25.4). These findings not only underscore the importance of all loss components across different scenarios but also highlight the critical role of confidence-based adaptive weighting, allowing the loss to emphasize the most reliable signals under varying conditions.

**T-EMA and Reset Strategy.** We evaluate the effectiveness of the T-EMA under dynamic domain shifts (CHiME3-Mix) by comparing it against two alternatives: (i) a *reset* variant that reinitializes CMA-ES parameters at the start of each utterance, and (ii) a variant that performs continuous adaptation without any resetting mechanism. As shown in Table 3, T-EMA consistently achieves lower WER than both variants. The *reset* variant yields the worst performance (26.5), indicating that discarding adaptation history prevents the accumulation of knowledge. Conversely, omitting reset entirely leads to sub-optimal results (25.4), suggesting that preserving historical information is important but needs to be regulated to avoid overfitting. T-EMA provides a principled balance

between stability and adaptability across utterances. The effectiveness of T-EMA is further analyzed with an increasing number of target domain samples in Appendix C.5.

## 6 Conclusions and Discussion

**Conclusions.** In this paper, we propose E-BATS, the first backpropagation-free test-time adaptation method for Speech Foundation Models that effectively balances adaptation accuracy and memory efficiency. E-BATS introduces a lightweight prompt adaptation module that directly adapts CNN-based feature encoders to mitigate acoustic domain shifts. A novel multi-scale loss function combining entropy minimization with utterance-level and token-wise feature alignment ensures fine-grained control over speech feature adaptation. Additionally, the test-time Exponential Moving Average mechanism stabilizes continuous adaptation in dynamic speech streams. Experimental results across four noisy datasets and diverse acoustic conditions demonstrate its superior performance, particularly in memory efficiency as the model size increases significantly.

**Limitations.** Although E-BATS is more theoretically efficient than other baseline methods through computation complexity comparision, which is reported in Appendix C.9, the iterative CMA-ES optimization introduces additional adaptation latency in practical environments. Specifically, the current implementation of CMA-ES does not fully exploit GPU parallelization, leading to sequential computation steps per utterance. While this latency is acceptable for scenarios without strict real-time requirements, it might pose challenges for latency-sensitive applications.

## Acknowledgments and Disclosure of Funding

This research was supported by The University of Melbourne's Research Computing Services and the Petascale Campus Initiative.

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

## A FID score of detailed domain shifts

To accurately quantify distributional shifts between the source and target domains, we employ the Fréchet Inception Distance (FID), a metric effective in capturing differences in both mean and covariance statistics of feature embeddings. Specifically, given the CNN-extracted latent embeddings $Z_t = [z_t^1, z_t^2, \ldots, z_t^{N_t}]$ for each utterance $X_t$, we first average these embeddings across all frames to obtain the utterance-level representation as $\frac{1}{N_t} \sum_{i=1}^{N_t} z_t^i \in \mathbb{R}^d$ due to the variable length of each utterance.

Considering the utterance-level embeddings from the source dataset $D_{\text{src}}$ and the target dataset $D_{\text{tgt}}$, we calculate their empirical mean vectors and covariance matrices as:

$$\boldsymbol{\mu}_{\text{src}} = \frac{1}{|D_{\text{src}}|} \sum_{X_t \in D_{\text{src}}} \frac{1}{N_t} \sum_{i=1}^{N_t} z_t^i, \quad \boldsymbol{\mu}_{\text{tgt}} = \frac{1}{|D_{\text{tgt}}|} \sum_{X_t \in D_{\text{tgt}}} \frac{1}{N_t} \sum_{i=1}^{N_t} z_t^i,$$

$$\Sigma_{\text{src}} = \frac{1}{|D_{\text{src}}| - 1} \sum_{X_t \in D_{\text{src}}} \left( \frac{1}{N_t} \sum_{i=1}^{N_t} z_t^i - \boldsymbol{\mu}_{\text{src}} \right) \left( \frac{1}{N_t} \sum_{i=1}^{N_t} z_t^i - \boldsymbol{\mu}_{\text{src}} \right)^{\top} + \varepsilon I,$$

$$\Sigma_{\text{tgt}} = \frac{1}{|D_{\text{tgt}}| - 1} \sum_{X_t \in D_{\text{tgt}}} \left( \frac{1}{N_t} \sum_{i=1}^{N_t} z_t^i - \boldsymbol{\mu}_{\text{tgt}} \right) \left( \frac{1}{N_t} \sum_{i=1}^{N_t} z_t^i - \boldsymbol{\mu}_{\text{tgt}} \right)^{\top} + \varepsilon I,$$

where $\varepsilon > 0$ is added to ensure numerical stability.

The final FID between the source and target domains is computed as:

$$\text{FID}(Overall) = \|\boldsymbol{\mu}_{\text{src}} - \boldsymbol{\mu}_{\text{tgt}}\|_2^2 + \text{Tr}\left( \Sigma_{\text{src}} + \Sigma_{\text{tgt}} - 2(\Sigma_{\text{src}}\Sigma_{\text{tgt}})^{\frac{1}{2}} \right),$$

where the first part is the Mean shift, and the second part is the Covariance shift. Mean Covariance shift Ratio will be calculated as $\frac{Mean\ shift}{Covariance\ shift}$.

Table 4: FID Scores and Mean Shift Ratios for Various Noisy Conditions

| Noisy Condition | Mean shift | Covariance shift | Mean/Covariance shift Ratio($\times$) |
|---|---|---|---|
| Gaussian noise | 0.031 | 0.004 | 7.8 |
| Single-domain environment noise (simulated) | 0.016 | 0.003 | 5.3 |
| Single-domain environment noise (real) | 0.029 | 0.004 | 7.3 |
| Mixed-domain environment noise | 0.019 | 0.005 | 3.8 |
| Speaking Style variability | 0.004 | 0.002 | 2.0 |

Table 5: FID Scores and Mean Shift Ratios for detailed Gaussian noise and single-domain environment noise

| Noisy Condition | Mean shift | Covaraince shift | Mean/Covariance shift Ratio ($\times$) |
|---|---|---|---|
| Gaussian noise $\sigma = 0.005$ | 0.017 | 0.003 | 5.7 |
| Gaussian noise $\sigma = 0.01$ | 0.028 | 0.003 | 9.3 |
| Gaussian noise $\sigma = 0.015$ | 0.036 | 0.004 | 9.0 |
| Gaussian noise $\sigma = 0.02$ | 0.042 | 0.004 | 10.5 |
| Cafe-real | 0.031 | 0.003 | 10.3 |
| Bus-real | 0.031 | 0.003 | 10.3 |
| Pedestain-real | 0.031 | 0.004 | 7.8 |
| Street-real | 0.022 | 0.004 | 5.5 |
| Cafe-simu | 0.021 | 0.003 | 7.0 |
| Bus-simu | 0.011 | 0.003 | 3.7 |
| Pedestain-simu | 0.017 | 0.004 | 4.3 |
| Street-simu | 0.015 | 0.003 | 5.0 |

# B Experiments

## B.1 Datasets

- **Gaussian Noise Data.** Following [16, 19], we corrupt the LibriSpeech [38] `test-other` split with zero-mean additive Gaussian noise to provide covariate shifts at different amplitudes ($\sigma \in \{0.0, 0.005, 0.01, 0.015, 0.02\}$). This setting provides a controlled evaluation of robustness to incremental noise severity.
- **Single-Domain Background Noise Data.**
    - **CHiME-3-single:** It is a noisy version of WSJ corpus with artificial and real-world environmental noises at 16 kHz. We utilize the official *simulated* and *real* enhanced evaluation sets from CHiME3 [39], which cover four challenging acoustic environments: bus, cafe, pedestrian area, and street junction. This setting simulates domain-specific, scene-consistent background conditions.
- **Multi-Domain and Wild Real-World Data.**
    - **CHiME-3-Mix:** All CHiME-3 scenarios are combined into a dynamic stream to simulate continuously shifting acoustic environments, similar to setups in continual test-time adaptation [17].
    - **CommonVoice (CV) [40]:** A crowdsourced project where volunteers contribute by reading Wikipedia sentences to produce 48 kHz audio samples. To align with the source ASR models' training conditions, we resampled these recordings to 16 kHz. The test set from the *en-June-22nd-2020* release was used to evaluate robustness against different speaking styles, accent variability, and crowd-sourced audio quality issues.
    - **TEDLIUM-v2 (TED):** Consists of oratory speech from TED conference videos with high quality stored at 16 kHz. We use the official test set for experiments, which introduces mismatches in recording quality and presentation style speech, diverging from the read speech in LibriSpeech or CommonVoice, and thus providing a natural domain shift. Following [16], transcripts across all datasets are converted to uppercase and stripped of punctuation, retaining only apostrophes.

## B.2 Baseline methods

The baseline methods include both BP TTA and BP-free TTAs.

**BP TTAs:**

- **TENT [13].** A fully test-time adaptation method that minimizes entropy by updating BatchNorm affine parameters online. We use it in episodic version since the batch size is small.
- **SAR [15].** A sharpness-aware and reliable entropy minimization method that selectively filters samples with large gradients and encourages the model weights to converge to a flat minimum, improving stability under wild domain shifts.
- **EATA [14].** An efficient TTA framework that selectively adapts on samples with lower uncertainty to reduce gradient noise and also mitigates catastrophic forgetting through a Fisher regularizer.
- **CoTTA [20].** A continuous TTA method that maintains a teacher-student adaptation strategy with stochastically restoring certain model parameters.
- **SUTA [16].** A single-utterance test-time adaptation method based on entropy minimization and minimum class confusion, adapted for CTC-based ASR.
- **CSUTA.** A continous version of SUTA with iteration step of 1, which is examined as one baseline in the work of [17].
- **DSUTA [17].** A dynamic variant of SUTA that adaptively resets or retains model updates based on domain shift detection with fast-slow adaptation strategy.
- **CEA [19].** Confidence-enhanced frame-level adaptation with short-term consistency regularization, proposed for wild acoustic test conditions.
- **SGEM [18].** A method leveraging beam-search logits and generalized entropy minimization for autoregressive ASR adaptation at sequence-level granularity.
- **AWMC [21].** A pseudo-labeling-based continual TTA algorithm for ASR that employs an anchor model, leader model and chaser model to achive stabel continous adaptation wihout forgetting.

**BP-free TTAs:**

- **LAME [33].** A training-free approach that corrects model outputs probabilities by estimating distribution drift in feature space.
- **T3A [27].** A TTA technique that adjusts classifier via pseudo-prototypes without requiring backward passes.
- **FOA [28].** A forward-only approach that optimizing learnable prompts with activation shifts to avoid forgetting issue and trivial solutions.

### B.3   Baseline methods hyperparameter setting

The detailed baseline settings and the hyperparameter tuning are presented for both BP TTA and BP-free TTA.

**BP TTA baslines**   The hyperparameter settings for TTA methods are organized as follows: For Speech Foundation Models (SFMs), we follow the configurations specified in the original papers for SUTA [16], DSUTA [17], CEA [19], and SGEM [18]. For CSUTA [17] and AWMC [21], which do not have released code, we adhere to the hyperparameters outlined in their official papers and the implementations from [17]. Additionally, we set the model to evaluation mode to maintain consistency.

For Visual Foundation Models (VFMs), to adapt the methods for SFMs with a batch size of 1 (BS=1), we follow the guidelines presented in [16, 19, 28]. All optimizers are configured to use the AdamW optimizer with the same learning rate as the TTA baseline methods for SFMs. Episodic methods are set with 10 iteration steps, while continuous methods use a single step. Minor adjustments are made for specific methods: for EATA [14], we use $e\_margin = 0.4 \times \ln(32)$ and $d\_margin = 1.0$; for SAR [15], $e\_margin = 0.4 \times \ln(32)$ and $reset\_constant = 0.3$; and for CoTTA [20], the augmentation threshold is set to 0.2, with augmentation limited to adding Gaussian noise.

**BP-free TTA baselines**   We follow the original hyperparameter settings for LAME [33] and T3A [27] as specified in their official papers. For FOA, the parameters are set as follows: $\sigma = 0.1$ (CMA-ES), $\alpha = 0.05$, and $\gamma = 0.1$.

In our approach, adaptation for each utterance is terminated early if the best fitness across iterations does not improve by at least 0.001 for three consecutive steps.

### B.4   Backbone models

Wav2Vec2ForCTC-Base [42] model employs a 12-layer Transformer encoder with a CNN-based feature extractor, representing lightweight ASR architectures optimized for fast inference. HuBERTForCTC-Large [43] model features a deeper 24-layer Transformer stack with a similar CNN front-end, offering a more powerful and robust ASR framework.

## C   Detailed Experiment results

### C.1   CHiME3

The detailed performance on CHiME3 dataset using Wav2Vec2ForCTC-Base and HuBERTForCTC-large are shown in Tables 6 and 7 respectively. It presents the performance comparison across four different acoustic environments, including cafe, bus, pedestrian, and street. For each acoustic condition, we also include the simulated and real-world noise conditions. The performance also demonstrated the superior performance of our approach over all BP-free TTA and most of BP-based TTA.

Table 6: Comparison of TTA methods across CHiME3-single (cafe, bus, pedestrian, street) by using Wav2vec2ForCTC-base model. WER in bold is the best performance within BP-free TTA methods, and the underlined WER is the best within both BP and BP-free TTA methods.

| Method | Cafe | | Bus | | Pedestrian | | Street | | Average | | |
|---|---|---|---|---|---|---|---|---|---|---|---|
| | Simu | Real | Simu | Real | Simu | Real | Simu | Real | Simu | Real | Overall |
| **Source** | 20.1 | 58.7 | 14.6 | 56.2 | 17.9 | 55.5 | 18.7 | 32.2 | 17.8 | 50.7 | 34.2 |
| *BP adaptation* | | | | | | | | | | | |
| **TENT (episodic)** | 20.0 | 58.4 | 14.5 | 55.9 | 17.8 | 55.2 | 18.7 | 32.0 | 17.8 | 50.4 | 34.1 |
| **EATA** | 20.0 | 55.3 | 14.9 | 54.1 | 18.3 | 51.6 | 18.9 | 31.4 | 18.0 | 48.1 | 33.1 |
| **SAR** | 19.6 | 56.2 | 14.4 | 56.5 | 17.8 | 52.8 | 18.7 | 33.0 | 17.6 | 49.6 | 33.6 |
| **CoTTA** | 19.2 | 58.6 | 14.2 | 49.2 | 16.8 | 55.4 | 17.8 | 32.1 | 17.0 | 48.8 | 16.2 |
| **CEA** | 17.3 | 45.8 | 13.1 | 41.8 | 16.1 | 38.0 | 17.1 | 25.2 | 15.9 | 37.7 | 26.8 |
| **SGEM** | 17.4 | 45.5 | 13.0 | 43.1 | 15.8 | 38.8 | 17.1 | 25.6 | 15.8 | 38.5 | 27.2 |
| **AWMC** | 19.5 | 62.9 | 14.5 | 54.6 | 17.4 | 39.0 | 18.5 | 31.9 | 17.5 | 50.6 | 34.0 |
| **SUTA** | 17.1 | 45.1 | 13.0 | 42.5 | 16.2 | 38.1 | 17.5 | 25.3 | 16.0 | 37.8 | 26.8 |
| **CSUTA (1 step)** | 17.7 | 40.3 | 14.8 | 41.1 | 16.8 | 36.4 | 18.7 | 26.0 | 17 | 36.0 | 26.5 |
| **DSUTA** | 16.4 | 36.3 | 13.0 | 39.6 | 15.2 | 32.8 | 15.6 | 22.4 | 15.1 | 32.8 | 24.0 |
| *BP-free adaptation* | | | | | | | | | | | |
| **T3A** | 20.9 | 61.3 | 15.3 | 59.0 | 18.5 | 58.4 | 19.6 | 33.6 | 18.6 | 53.1 | 35.9 |
| **LAME** | 20.9 | 61.6 | 15.2 | 58.9 | 18.6 | 58.9 | 19.8 | 33.9 | 18.6 | 53.3 | 36.0 |
| **FOA** | 19.9 | 52.5 | 14.6 | 50.5 | 18.0 | 48.1 | 18.6 | 31.0 | 17.8 | 45.5 | 31.7 |
| **Ours** | **16.1** | **37.9** | **13.1** | **37.4** | **15.1** | **33.1** | **15.4** | **24.0** | **14.9** | **33.1** | **24.0** |

Table 7: Comparison of TTA methods across CHiME3-single (cafe, bus, pedestrian, street) by using HuBERTForCTC-large. WER in bold is the best performance within BP-free TTA methods, and the underlined WER is the best within both BP and BP-free TTA methods.

| Method | Cafe | | Bus | | Pedestrian | | Street | | Average | | |
|---|---|---|---|---|---|---|---|---|---|---|---|
| | Simu | Real | Simu | Real | Simu | Real | Simu | Real | Simu | Real | Overall |
| **Source** | 9.2 | 27.9 | 8.5 | 26.2 | 9.2 | 24.3 | 10.1 | 16.6 | 9.3 | 23.8 | 16.6 |
| *BP adaptation* | | | | | | | | | | | |
| **TENT (episodic)** | 9.2 | 27.5 | 8.5 | 25.8 | 9.2 | 24.0 | 10.0 | 16.5 | 9.2 | 23.5 | 16.4 |
| **EATA** | 9.2 | 27.0 | 8.7 | 25.8 | 9.2 | 23.2 | 9.9 | 16.4 | 9.3 | 23.1 | 16.2 |
| **SAR** | 9.1 | 28.0 | 8.4 | 26.1 | 9.1 | 23.2 | 9.9 | 17.3 | 9.1 | 23.7 | 16.4 |
| **CoTTA** | 9.3 | 26.8 | 8.5 | 25.5 | 9.1 | 23.9 | 9.9 | 16.4 | 9.2 | 23.2 | 16.2 |
| **CEA** | 8.6 | 22.5 | 8.1 | 22.0 | 8.3 | 19.6 | 9.2 | 14.7 | 8.6 | 19.7 | 14.2 |
| **SGEM** | 8.7 | 22.5 | 8.1 | 21.8 | 8.5 | 19.9 | 9.5 | 14.7 | 8.6 | 19.7 | 14.2 |
| **AWMC** | 9.5 | 25.5 | 8.9 | 23.9 | 9.4 | 22.5 | 10.4 | 16.5 | 9.6 | 22.1 | 15.9 |
| **SUTA** | 8.6 | 22.9 | 8.1 | 22.2 | 8.4 | 19.6 | 9.4 | 14.6 | 8.6 | 19.8 | 14.2 |
| **CSUTA (1 step)** | 9.3 | 22.8 | 8.9 | 22.2 | 9.5 | 19.7 | 9.9 | 14.7 | 9.4 | 19.9 | 14.7 |
| **DSUTA** | 8.7 | 20.3 | 8.3 | 20.0 | 8.6 | 17.5 | 9.3 | 13.9 | 8.7 | 17.9 | 13.3 |
| *BP-free adaptation* | | | | | | | | | | | |
| **T3A** | 18.2 | 42.1 | 18.2 | 42.1 | 17.3 | 37.5 | 18.2 | 29.0 | 18.0 | 37.7 | 27.9 |
| **LAME** | 9.7 | 30.2 | 8.9 | 27.9 | 9.6 | 25.4 | 10.6 | 17.8 | 9.7 | 25.3 | 17.5 |
| **FOA** | 9.6 | 26.9 | 8.6 | 26.2 | 9.3 | 23.4 | 10.4 | 16.4 | 9.5 | 23.2 | 16.4 |
| **Ours** | **9.2** | **20.7** | **8.4** | **21.3** | **9.2** | **18.9** | **10.0** | **14.2** | **9.2** | **18.8** | **14.0** |

## C.2 Ablation study with multiple seeds

We repeat each ablation experiment using three different random seeds in Figure 8 to ensure that our results are robust and not the artifact of any single initialization.

Table 8: Ablation study with mean±std WER (%) over 3 seeds

| Prompt Adaptation | | | Loss Function (w/ T-EMA) | | | | Loss Function (w/o T-EMA) | | | | T-EMA Mechanism | | |
|---|---|---|---|---|---|---|---|---|---|---|---|---|---|
| Feat | Trans | WER | $L_{ent}$ | $L_{utt}$ | $L_{token}$ | WER | $L_{ent}$ | $L_{utt}$ | $L_{token}$ | WER | T-EMA | Reset | WER |
| ✓ | — | 24.0±0.0 | ✓ | ✓ | ✓ | 24.0±0.0 | ✓ | ✓ | ✓ | 25.4±0.0 | ✓ | — | 24.3±0.0 |
| — | ✓ | 34.1±0.1 | ✓ | ✓ | — | 24.2±0.1 | ✓ | ✓ | — | 25.5±0.1 | — | ✓ | 26.5±0.0 |
| — | — | — | ✓ | — | — | 24.6±0.1 | ✓ | — | — | 64.3±11.5 | — | — | 25.4±0.0 |

## C.3 Sensitive Analysis

### C.3.1 Sensitivity of T-EMA Decay Parameter

Table 9 below presents our analysis on the EMA parameter $\gamma$, where we evaluated performance across $\gamma \in \{0.50, 0.60, 0.70, 0.80, 0.90, 0.95, 0.99\}$ under the caf-real condition. We observed a clear U-shaped WER curve: performance improves up to $\gamma = 0.90$ (best WER 37.9%), and then degrades as $\gamma$ increases further. This behavior aligns with our expectation and is also observed in other datasets that smaller $\gamma$ values lead to unstable, overly reactive updates, while larger values result in over-smoothing and hinder adaptation.

Table 9: WER (%) of Wav2Vec2ForCTC on caf-real CHiME-3 for different T-EMA decay $\gamma$.

| $\gamma$ | 0.50 | 0.60 | 0.70 | 0.80 | 0.90 | 0.95 | 0.99 |
|---|---|---|---|---|---|---|---|
| WER (%) | 39.1 | 39.1 | 38.4 | 38.1 | 37.9 | 38.3 | 39.9 |

### C.3.2 Sensitivity of Loss Component Weights

To further understand how the relative weighting of loss components affects the CMA-ES optimization process, we conducted a sensitivity analysis of the loss weights $\alpha$ and $\beta$ for the entropy loss and utterance-level loss, which in turn leads to changes in the token-level weight $c$ (Minmax normalization Equation in Section 3.3). The WER results using the caf-real condition of the CHiME3 dataset are shown in Table 10 below. The results reveal that WER is remarkably stable across a wide range of weight settings. WER stays within a narrow range across different values of $\alpha$ and $\beta$, showing a broad area of near-optimal performance instead of a single best point. The results suggest that both components of the loss must be balanced, as extreme values for either can hurt performance. The chosen setting ($\alpha = 1.0$, $\beta = 2.0$) sits comfortably in this stable region and consistently yields the best performance.

Table 10: WER (%) of Wav2Vec2ForCTC on caf-real CHiME-3 under varying $\alpha$ (rows) and $\beta$ (columns), representing varying weights for the token-level loss.

| $\alpha \setminus \beta$ | 0.5 | 1.0 | 1.5 | 2.0 | 2.5 | 3.0 | 3.5 | 4.0 | 4.5 | 5.0 |
|---|---|---|---|---|---|---|---|---|---|---|
| 0.5 | 37.7 | 38.2 | 38.5 | 38.6 | 39.1 | 39.4 | 39.2 | 39.8 | 39.4 | 39.7 |
| 1.0 | 38.1 | 38.1 | 38.0 | 37.9 | 38.4 | 38.3 | 38.4 | 38.8 | 38.7 | 39.6 |
| 1.5 | 38.5 | 38.1 | 38.4 | 38.1 | 38.5 | 38.0 | 38.2 | 37.8 | 38.3 | 38.4 |
| 2.0 | 38.4 | 38.3 | 38.0 | 37.9 | 37.9 | 38.1 | 38.0 | 38.3 | 38.0 | 38.4 |

### C.3.3 Sensitivity of CMA-ES to Target Domain Complexity

We selected four target domains exhibiting increasing variability, quantified by the covariance shift offered by Fréchet Inception Distance (FID) of the embedding distribution between the source and target. These range from low to high variability: Gaussian noise ($\sigma = 0.01$), CHiME-3 cafe-simulated, CHiME-3 cafe-real, and CHiME-3 mixed. As shown in Table 11 below, with CMA-ES

population size increasing from 10 to 100, the largest gains occur moving from 10 to 40 candidates across all domains. Beyond 50 candidates, performance plateaus. These results show that while more complex domains (cafe-real, mixed) begin at higher absolute WER, they exhibit the same early-saturation behavior as simpler conditions. Therefore, sensitivity to domain complexity may not be directly related to the number of prompt candidates considered.

Table 11: WER (%) across different population sizes on four target domains with varying complexity.

| Candidate Size | 10 | 20 | 30 | 40 | 50 | 60 | 70 | 80 | 90 | 100 |
|---|---|---|---|---|---|---|---|---|---|---|
| **Gaussian ($\sigma = 0.01$)** | 16.2 | 15.6 | 15.2 | 14.9 | 14.8 | 14.8 | 14.7 | 14.7 | 14.7 | 14.7 |
| **Cafe-simulated** | 17.0 | 16.5 | 16.5 | 16.2 | 16.1 | 16.3 | 16.1 | 16.0 | 16.0 | 16.2 |
| **Cafe-real** | 41.8 | 39.3 | 38.3 | 38.3 | 37.9 | 38.0 | 37.9 | 37.9 | 38.0 | 37.8 |
| **CHiME3 Mixed** | 25.7 | 25.1 | 24.7 | 24.6 | 24.3 | 24.3 | 24.4 | 24.2 | 24.4 | 24.3 |

### C.3.4 Sensitivity of CMA-ES to Candidate Size and Iteration Steps

We find the expected trade-off between the population size and optimization iterations with word error rate (WER) (Table 12 and Table 13 below). As shown in Table 12 (caf-real condition of CHiME-3 using Wav2Vec2ForCTC), increasing the CMA-ES candidate size from 10 to 40 yields the most significant gains. Table 13 demonstrates that moving from 5 to 25 iterations captures most of the benefit. We observe the same early-saturation behavior across other conditions (e.g., bus-real, street-real) and datasets.

Table 12: WER (%) of Wav2Vec2ForCTC on the *caf-real* condition of CHiME-3 under different CMA-ES population sizes (iteration steps = 25).

| Candidate Size | 10 | 20 | 30 | 40 | 50 | 60 | 70 | 80 | 90 | 100 |
|---|---|---|---|---|---|---|---|---|---|---|
| **WER (%)** | 41.8 | 39.3 | 38.3 | 38.3 | 37.9 | 38.0 | 37.9 | 37.9 | 38.0 | 37.8 |

Table 13: WER (%) of Wav2Vec2ForCTC on the *caf-real* condition of CHiME-3 under different CMA-ES iteration steps (population size = 50).

| Iteration Steps | 5 | 10 | 15 | 20 | 25 | 30 | 35 | 40 | 45 | 50 |
|---|---|---|---|---|---|---|---|---|---|---|
| **WER (%)** | 40.8 | 39.9 | 38.6 | 38.0 | 37.9 | 38.0 | 37.9 | 37.6 | 38.4 | 38.0 |

### C.4 Impact of Utterance-level Loss on Trivial Predictions

We analyzed and compared the occurrence of trivial predictions under two conditions: (i) entropy-only and (ii) entropy with utterance-level loss, using the CHiME-3-Mix dataset. As shown in Table 14, we observed two types of trivial solutions: blank predictions and single-character predictions (entropy = 0 with only one character predicted). It is evident that the utterance-level loss significantly reduces the incidence of trivial solutions from 37.5% to 0.61%, thereby improving the WER.

### C.5 Effectiveness of Continual Adaptation with T-EMA

We reported the WER against the number of process utterances under two variations of the continual adaptation process: (i) T-EMA with resetting, reported in Table 3 in the ablation study; and (ii) T-EMA with exponential averaging (proposed), presented in Table 3 and Section 3.4. We evaluated their performance across varying numbers of target samples from 100 to 800 in Table 15 below, using the LibriSpeech dataset with Gaussian noise ($\sigma = 0.015$). With the resetting approach, performance increases up to 300 samples and then declines with additional samples, suggesting that resetting CMA-ES prevents leveraging prior adaptation. In contrast, the proposed exponential averaging method demonstrates relatively stable performance, indicating that even with a small number of samples, reliable adaptation can be achieved. This is likely due to the T-EMA mechanism accumulating knowledge from earlier utterances and updating CMA-ES in a more favorable region

Table 14: Trivial solutions on CHiME-3-Mix: number and percentage of trivial predictions among all predictions.

| Configuration | Blank | Single-char | Total problematic | WER (%) |
|---|---|---|---|---|
| **Entropy-only** | 42 (1.59%) | 948 (35.91%) | 990 (37.50%) | 49.6 |
| **+ Utterance-level loss** | 12 (0.45%) | 4 (0.15%) | 16 (0.61%) | 25.5 |

Table 15: WER (%) against the number of processed utterances with T-EMA (resetting) and T-EMA (proposed) under the LibriSpeech dataset with Gaussian noise ($\sigma = 0.015$).

| Method | 100 | 200 | 300 | 400 | 500 | 600 | 700 | 800 |
|---|---|---|---|---|---|---|---|---|
| **T-EMA (Resetting)** | 20.4 | 19.7 | 19.8 | 20.4 | 20.8 | 20.8 | 21.1 | 21.3 |
| **T-EMA (Proposed)** | 17.9 | 17.0 | 17.2 | 17.4 | 17.6 | 17.5 | 17.8 | 17.7 |
| **Performance Gain** | 2.5 | 2.7 | 2.6 | 3.0 | 3.2 | 3.3 | 3.3 | 3.6 |

of the search space. The improved performance over the resetting mechanism further demonstrates that our method is robust to the number of samples used for adaptation, and that the proposed T-EMA effectively stabilizes the adaptation process.

## C.6 Memory Usage

Figure 6 shows the memory usage comparison between two SFMs: Wav2Vec2ForCTC-Base and HuBERTForCTC-Large. The increasing model size leads to a significant increase in memory usage for adaptation, especially for the BP-based TTA, whereas our method demonstrates only a slight increase in memory usage.

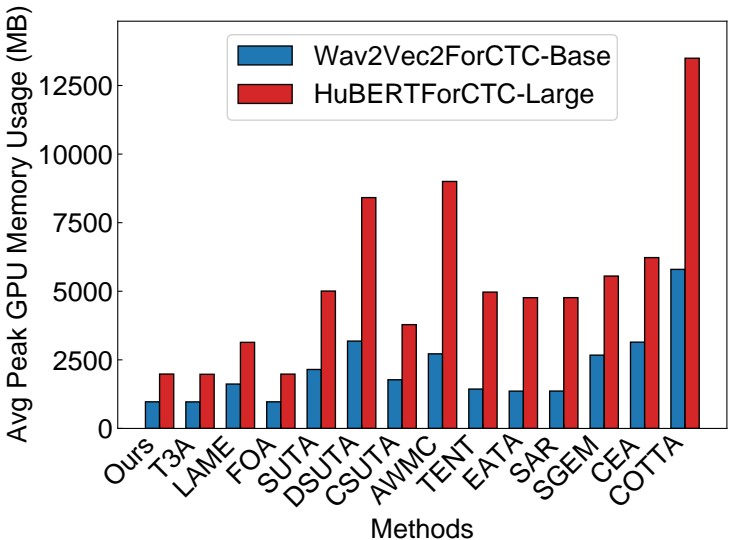

Figure 6: Comparison of Average Peak GPU memory usage of different TTA methods across all datasets with two different scale backbones.

## C.7 Controlling Prompt Vector Dimensionality for CMA-ES Efficiency

The computational cost of CMA-ES with increasing dimensionality $d$ may cause concerns. Indeed, the full-rank CMA-ES algorithm has a per-iteration complexity of $\mathcal{O}(Jd^2 + d^3)$, where $J$ is the number of sampled solutions (i.e., prompts in our case). To mitigate this cost, we design our prompt vector to have a fixed dimensionality of $512$, aligned with the embedding size commonly used in speech foundation models, ensuring that $d$ remains tractable in practical scenarios. Furthermore, as

Table 16: WER (%) of Wav2Vec2ForCTC on the *caf-real* condition of CHiME-3 under different CMA-ES candidate sizes $J$ (iteration steps = 25).

| $J$ | 10 | 20 | 30 | 40 | 50 | 60 | 70 | 80 | 90 | 100 |
|---|---|---|---|---|---|---|---|---|---|---|
| **WER (%)** | 41.8 | 39.3 | 38.3 | 38.3 | 37.9 | 38.0 | 37.9 | 37.9 | 38.0 | 37.8 |

Table 17: Per-utterance adaptation time complexity (big-$\mathcal{O}$) for E-BATS and top TTA baselines.

| Method | Time Complexity per Utterance |
|---|---|
| **DSUTA** | $\mathcal{O}(N \times (F + E + P))$ |
| **FOA** | $\mathcal{O}(N \times [J(F + E + Ld^2) + d^3])$ |
| **E-BATS** | $\mathcal{O}(N \times [J(F + E + L + d + L|V|d) + d^3])$ |

shown in Table 16, with our grid search of $J$, we observe the optimal $J$ always much smaller than $d$; in our configuration, we fix $J = 50$, which remains significantly smaller than $d$. This ensures that the asymptotic complexity does not exceed $\mathcal{O}(d^3)$. Thus, while CMA-ES theoretically scales cubically with $d$, our design choices effectively cap the computational overhead in the context of high-dimensional but fixed-size prompt vectors.

### C.8 Adaptation performance on Speech Emotion Recognition Task

To further test the generality of our method beyond ASR with CTC, we applied E-BATS to speech emotion recognition tasks (cross-entropy loss) using the IEMOCAP dataset and the SpeechBrain/emotion-recognition-wav2vec2-IEMOCAP model under additive Gaussian noise ($\sigma = 0.02$). Adaptation increased emotion prediction accuracy from 38.8% to 43.4%, demonstrating that our E-BATS framework can enhance the performance of other downstream tasks using SFMs.

### C.9 Adaptation Speed and Computation Complexity

We compute the per-utterance time complexity to compare E-BATS with the top-performing backpropagation-based (DSUTA) and backpropagation-free (FOA) methods (see Table 17). DSUTA scales with the large number of model parameters via backpropagation ($P$), often in hundreds of millions, whereas E-BATS only involves $\mathcal{O}(d^2)$ and $\mathcal{O}(d^3)$ operations (with $d = 512$). Since $P \geq d^3$ in most SFM models, E-BATS achieves significant efficiency gains over DSUTA. While both FOA and E-BATS share the same asymptotic complexity of $\mathcal{O}(d^3)$ for CMA-ES updates, E-BATS is practically more efficient since FOA requires a $3^3$ multiplicative factor of $\mathcal{O}(d^3)$ as it needs three prompts for adaptation. FOA also incurs an additional $\mathcal{O}(Ld^2)$ cost (over $d$) per prompt for prompt attention with $L$ transformer layer encoders. Nonetheless, our main focus, as noted in the introduction, is the trade-off between accuracy and memory efficiency, which is more critical for resource-constrained devices.

where:

- $F$: cost of one forward pass
- $E$: cost of entropy loss calculation
- $N$: number of adaptation iteration steps per utterance
- $J$: number of candidate prompts per iteration
- $|V|$: size of the token class

## D Algorithms

The algorithm for LPA per utterance and for T-EMA is shown in Algorithm 1 and Algorithm 2 respectively.

---

**Algorithm 1** Lightweight Prompt Adaption (Per Utterance)

---

**Require:** CMA-ES params $\phi_t^0$, max steps $K$, Utterance $\boldsymbol{X}_t$
**Ensure:** Adapted predictions $\hat{\boldsymbol{y}}$
 1: best_loss $\leftarrow \infty$
 2: **for** $k = 1$ to $K$ **do**
 3:    Sample prompts $S_t^k = \left[\boldsymbol{s}_{t,1}^k, \boldsymbol{s}_{t,2}^k, \ldots, \boldsymbol{s}_{t,J}^k\right]$ from CMA-ES $\phi_{t,k-1}$
 4:    Inject $\boldsymbol{s}_{t,j}^k$ with $\boldsymbol{Z}_t$ and feedforward pass
 5:    Compute loss $\boldsymbol{L}_{adapt,all}^k = \left[\boldsymbol{L}_{adapt,1}^k, \boldsymbol{L}_{adapt,2}^k, \ldots, \boldsymbol{L}_{adapt,J}^k\right]$
 6:    **if** $\min_{j \in \{1,\ldots,J\}} \boldsymbol{L}_{adapt,j}^k <$ best_loss **then**
 7:      best_loss $\leftarrow \boldsymbol{L}_{adapt,j}^k$
 8:      $\hat{\boldsymbol{Y}}_t \leftarrow$ decode adapted output with $\boldsymbol{s}_{t,j}^k$
 9:    **end if**
10:    $\phi_t^k \leftarrow \text{Update}(\phi_t^{k-1}, \boldsymbol{L}_{adapt,all}^k, \boldsymbol{S}_t^k)$
11: **end for**

---

---

**Algorithm 2** T-EMA Updating Strategy

---

**Require:** Utterance data $T$, CMA-ES params $\phi_0^0$, Iteration Steps $K$
 1: Initialize $\phi_{ema} = \{C_{ema}, m_{ema}, \sigma_{ema}\} \leftarrow \phi_0^0$
 2: **for** each utterance $t$ in $T$ **do**
 3:    $\phi_t^0 \leftarrow \phi_t^K$ Run Algorithm 1
 4:    EMA Update $(\phi_{ema}, \phi_t^K)$
 5:    $\phi_{t+1}^0 \leftarrow \phi_{ema}$
 6: **end for**

---

# E   Ethical Consideration

Our research fully complies with the NeurIPS Code of Ethics. We exclusively utilize publicly available datasets, pretrained models, baseline methods, and their accompanying code, strictly adhering to their respective licenses and usage protocols. We did not collect any new data, nor do our adaptation methods pose privacy risks or enable misuse. Thus, our work does not introduce broader negative societal impacts, eliminating the need for additional safeguards beyond standard ethical research practices.

Moreover, our method has potential positive societal impacts, including improving the accessibility and reliability of speech recognition technology in noisy real-world environments, thereby benefiting communication technologies, assistive systems, and applications serving diverse and inclusive user populations.

