# OpenReview forum: "E-BATS: Efficient Backpropagation-Free Test-Time Adaptation for Speech Foundation Models"
_NeurIPS.cc/2025/Conference — NeurIPS 2025 poster_

### Official Review · Reviewer_LXCm · 2025-06-03

**Clarity:** 3
**Significance:** 3
**Originality:** 3
**Rating:** 4
**Confidence:** 2

**Summary:**

E-BATS is a backpropagation-free Test-Time Adaptation (TTA) method developed for Speech Foundation Models (SFMs). It is designed to adapt effectively to domain shifts that commonly occur in real-world scenarios, such as background noise, speaker accent, and device variation, while also minimizing memory usage. The method inserts prompt vectors into a CNN-based feature encoder, allowing the adjustment of the feature space without changing the model weights. These prompts are optimized without backpropagation using Covariance Matrix Adaptation Evolution Strategy (CMA-ES). To further enhance performance, the authors employ entropy minimization, a multi-scale loss based on both utterance-level and token-level alignment, and a technique called T-EMA. The approach shows strong performance compared to various benchmarks across four datasets and two different models.

**Questions:**

I wrote my questions in weaknesses.

**Ethical Concerns:**

["NO or VERY MINOR ethics concerns only"]

**Final Justification:**

Given that my score was already on the positive side, I will retain it as is.

**Limitations:**

yes

**Paper Formatting Concerns:**

Nothing.

**Quality:**

2

**Strengths And Weaknesses:**

1. Strengths

- The authors are the first to propose a backpropagation-free test-time adaptation (TTA) method specifically for speech foundation models.

- They support their claims through extensive comparisons across various models, datasets, and methods, as well as comprehensive ablation studies.

- In addition, the figures presented in the paper are clear and intuitive, making the overall methodology easier to understand.

2. Weaknesses

- The CMA-ES method appears to rely on the covariance matrix, which raises the question of whether the computational cost becomes significant as the dimensionality $d$ increases.

- While the overall results in the paper focus on speech recognition, speech foundation models are typically used across a wide range of downstream tasks beyond just ASR. Including results on additional tasks would make the method appear more broadly applicable and universal.

- As someone less familiar with backpropagation-free TTA methods, I am also curious about the temporal dynamics of speech. Since speech signals often exhibit varying characteristics across frames, would it be possible to design the prompts to vary across frames as well, rather than keeping them fixed?

---

> ### Author Rebuttal · Authors · 2025-07-30
>
> ## Response to Reviewer
> We thank the reviewer for the detailed feedback. We greatly appreciate the reviewer's recognition of our work as offering the "first" backpropagation-free Test‑Time Adaptation framework tailored for speech foundation models with "strong performance" and "extensive comparisons". Below, we address the reviewer's concerns regarding the computational cost of CMA-ES with dimensionality *d*, applicability beyond ASR, and the applicability of designing prompts to vary across frames. We will include these results and analysis in the appendix of the revised version.
>
> ---
>
> - **W1:** The CMA-ES method appears to rely on the covariance matrix, which raises the question of whether the computational cost becomes significant as the dimensionality *d* increases.
>
>   **Response:** We appreciate the reviewer’s concern regarding the computational cost of CMA-ES with increasing dimensionality *d*. Indeed, the full-rank CMA-ES algorithm has a per-iteration complexity of *O*(*J* *d*² + *d*³), where *J* is the number of sampled solutions (i.e., prompts in our case). To mitigate this cost, we design our prompt vector to have a fixed dimensionality of 512, aligned with the embedding size commonly used in speech foundation models, ensuring that *d* remains tractable in practical scenarios. Furthermore, As shown in table 1, with our grid search of J, we observe the optimal J always much smaller than d, in our configuration, we fix J=50, which remains significantly smaller than *d*. This ensures that the asymptotic complexity does not exceed *O*(*d*³). Thus, while CMA-ES theoretically scales cubically with *d*, our design choices effectively cap the computational overhead in the context of high-dimensional but fixed-size prompt vectors.
>
>   **Table 1.** WER (%) of Wav2Vec2ForCTC on the _caf-real_ condition of CHiME-3 under different CMA-ES candidate sizes *J*
>   _Iteration steps = 25_
>
>   | *J*   | 10   | 20   | 30   | 40   | 50   | 60   | 70   | 80   | 90   | 100  |
>   |-----|------|------|------|------|------|------|------|------|------|------|
>   | WER (%) | 41.8 | 39.3 | 38.3 | 38.3 | 37.9 | 38.0 | 37.9 | 37.9 | 38.0 | 37.8 |
>
> ---
>
> - **W2:** While the overall results in the paper focus on speech recognition, speech foundation models are typically used across a wide range of downstream tasks beyond just ASR. Including results on additional tasks would make the method appear more broadly applicable and universal.
>
>   **Response:** We further applied E-BATS to speech emotion recognition using the IEMOCAP benchmarking dataset and the `SpeechBrain/emotion-recognition-wav2vec2-IEMOCAP` model. Emotion prediction accuracy improved from 38.8% (source model without adaptation) to 43.4% (with E-BATS) under additive Gaussian noise (σ = 0.02). This shows that our E-BATS framework can generalize to other speech tasks. We will include these experiments in the appendix of the revised version.
>
> ---
>
> - **W3:** As someone less familiar with backpropagation-free TTA methods, I am also curious about the temporal dynamics of speech. Since speech signals often exhibit varying characteristics across frames, would it be possible to design the prompts to vary across frames as well, rather than keeping them fixed?
>
>   **Response:** This is an interesting point raised. While we recognize that speech signals exhibit varying characteristics over time, especially across extended durations, our current approach is based on the practical assumption, common in domain adaptation for speech, that distribution shifts (such as speaker accent or background noise) tend to affect an utterence relatively uniformly. For example, a speaker’s accent or background noise typically remains stable over a typical 10-second utterance, resulting in a coherent distribution shift within that utterence. This justifies our use of a global prompt.
>
>   That said, we also acknowledge that learning frame-wise prompts could potentially capture finer-grained variations, but this is technically challenging: speech utterances naturally vary in length, so adopting frame-wise prompts would require padding or truncating to a uniform length and flattening a much larger prompt tensor for optimization (e.g., with CMA-ES), which substantially increases complexity and decreases optimization stability. Nevertheless, varying the frame-wise prompt remains an interesting direction and will be considered in the future.
>
> ---
>
> We appreciate your feedback and are happy to provide further clarification if needed.

---

### Official Review · Reviewer_4Yrj · 2025-07-02

**Clarity:** 3
**Significance:** 3
**Originality:** 3
**Rating:** 5
**Confidence:** 2

**Summary:**

This paper introduces E-BATS, a novel backpropagation-free framework for efficiently adapting speech foundation models to new, noisy environments at test-time. By optimizing a lightweight "prompt" vector using a forward-pass-only process, the method avoids the high memory costs of traditional backpropagation-based adaptation. Experiments demonstrate that E-BATS achieves strong accuracy, comparable to or exceeding state-of-the-art methods, while using significantly less memory.

**Questions:**

Regarding the continual adaptation process, it would be useful to see how performance evolves as more test samples from a target domain are processed. For example, a plot of WER against the number of processed utterances could help illustrate the adaptation dynamics over time.

**Ethical Concerns:**

["NO or VERY MINOR ethics concerns only"]

**Final Justification:**

My concerns have been well addressed. I have increased my score to 5.

**Limitations:**

yes

**Quality:**

3

**Strengths And Weaknesses:**

## Strengths
1. The proposed method was evaluated on two speech foundation models (Wav2Vec2 and Hubert) across several ASR domain shift scenarios. The experiments show the method's effectiveness under the tested conditions.

2. The paper is clearly written and the methodology is easy to follow. The inclusion of an ablation study helps clarify the contribution of individual components of the proposed framework.

## Weaknesses

1. The evaluation is currently limited to the ASR task. Exploring the method's applicability to other speech tasks (e.g. emotion recognition, speaker verification) is necessary since Wav2Vec2 and Hubert are general-purpose speech foundation models.

2. Based on the ablation study, the inclusion of the $L_{utt}$ and $L_{token}$ loss components provides a marginal improvement when the T-EMA module is active.

---

> ### Author Rebuttal · Authors · 2025-07-30
>
> ## Response to Reviewer
>
> We thank the reviewer for the detailed feedback. We greatly appreciate the reviewer's recognition of our work as offering a "novel" backpropagation-free Test‑Time Adaptation framework tailored to real‑world acoustic shifts with "strong accuracy" and "using significantly less memory". Below, we address the reviewer's concerns regarding applicability beyond ASR, the role of our loss components, and adaptation dynamics over time. We will include these results and analysis in the appendix of the revised version.
>
> ---
> - **W1:** The evaluation is currently limited to the ASR task. Exploring the method's applicability to other speech tasks (e.g. emotion recognition, speaker verification) is necessary since Wav2Vec2 and Hubert are general-purpose speech foundation models.
>
>   **Response:** We further applied E-BATS to speech emotion recognition using the IEMOCAP benchmarking dataset and the `SpeechBrain/emotion-recognition-wav2vec2-IEMOCAP` model. Emotion prediction accuracy improved from 38.8% (source model without adaptation) to 43.4% (with E-BATS) under additive Gaussian noise (σ = 0.02). This shows that our E-BATS framework can generalize to other speech tasks. We will include these experiments in the appendix of the revised version.
>
> ---
> - **W2:** Based on the ablation study, the inclusion of the $L_{utt}$ and $L_{token}$ loss components provides a marginal improvement when the T-EMA module is active.
>
>   **Response:** T-EMA is designed to maintain stability during the adaptation process by preventing drastic changes caused by individual utterances, resulting in more gradual and reliable model updates. With T-EMA applied, both utterance-level and token-level adaptations benefit from this enhanced stability, making them unlikely to be destabilized by any single utterance. As a result, utterance-level or token-level adaptation on top of T-EMA yields slight improvements. In contrast, without T-EMA, adaptation based on entropy loss alone (49.6%) can be significantly affected by inaccuracies from a single utterance. In such cases, utterance-level and token-level losses play a crucial stabilizing role (25.4%). Overall, these results further validates the reliability of utterance-level and token-level losses. We believe this provides valuable insights into the impact of different loss designs under various settings for SFMs, and we will clarify this further in the ablation study subsection.
>
> ---
>
> - **Q1:** Regarding the continual adaptation process, it would be useful to see how performance evolves as more test samples from a target domain are processed. For example, a plot of WER against the number of processed utterances could help illustrate the adaptation dynamics over time.
>
>   **Response:** We reported the WER against the number of process utterances under two variations of the continual adaptation process: (i) T-EMA with resetting, reported in Table 3 in the ablation study; and (ii) T-EMA with exponential averaging (proposed), presented in Table 3 and Section 3.4. We evaluated their performance across varying numbers of samples from 100 to 800 in Table 1 below, using the LibriSpeech dataset with Gaussian noise (σ = 0.015). With the resetting approach, performance increases up to 300 samples and then declines with additional samples, suggesting that resetting CMA‑ES prevents leveraging prior adaptation. In contrast, the proposed exponential averaging method demonstrates relatively stable performance, indicating that even with a small number of samples, reliable adaptation can be achieved. This is likely due to the T-EMA mechanism accumulating knowledge from earlier utterances and updating CMA‑ES in a more favorable region of the search space. The improved performance over the resetting mechanism further demonstrates that our method is robust to the number of samples used for adaptation, and that the proposed T-EMA effectively stabilizes the adaptation process. We will include the results in the appendix.
>
>   **Table 1.** WER against the number of processed utterances with T-EMA (resetting) and T-EMA (proposed).
>
>   | Method                | 100  | 200  | 300  | 400  | 500  | 600  | 700  | 800  |
>   |-----------------------|------|------|------|------|------|------|------|------|
>   | T‑EMA (Resetting)     | 20.4 | 19.7 | 19.8 | 20.4 | 20.8 | 20.8 | 21.1 | 21.3 |
>   | T‑EMA (Proposed)      | 17.9 | 17.0 | 17.2 | 17.4 | 17.6 | 17.5 | 17.8 | 17.7 |
>   | **Performance Gain**  | 2.5  | 2.7  | 2.6  | 3.0  | 3.2  | 3.3  | 3.3  | 3.6  |
>
> ---
>
> We appreciate your time and welcome any further questions or suggestions.

---

> > ### Comment · Reviewer_4Yrj · 2025-08-06
> >
> > Thanks for your response. My concerns have been well addressed. I have increased my score to 5.

---

### Official Review · Reviewer_Wcum · 2025-07-03

**Clarity:** 3
**Significance:** 3
**Originality:** 3
**Rating:** 4
**Confidence:** 2

**Summary:**

This paper addresses a critical limitation of speech foundation models—their susceptibility to performance degradation under real-world acoustic domain shifts such as background noise and speaker accents. The authors explore Test-Time Adaptation (TTA) as a strategy to overcome such challenges without requiring access to source data or labels. The primary innovation is E-BATS, a backpropagation-free TTA framework tailored specifically for speech tasks.

**Questions:**

N/A

**Ethical Concerns:**

["NO or VERY MINOR ethics concerns only"]

**Quality:**

3

**Strengths And Weaknesses:**

Strengths:

1. The paper identifies a clear gap in existing TTA methods—most are either memory-intensive (due to backpropagation) or ineffective (being designed for vision tasks). E-BATS provides a compelling, efficient, and accurate alternative for speech scenarios.

2. The method integrates:
o Prompt-based adaptation, enabling forward-only updates that significantly reduce computational overhead.
o Multi-scale loss, which captures both global (utterance-level) and local (token-level) distributional shifts, often overlooked in vision-
based approaches.
o Exponential Moving Average (EMA), which ensures smoother and more stable adaptation across test utterances.

3. The framework demonstrates strong generalization across four noisy speech datasets and sixteen acoustic domains, showing consistent and meaningful improvements.

Weaknesses:

1. The proposed multi-scale loss combines utterance-level and token-level objectives, but it remains unclear how the relative weighting of these loss components affects the CMA-ES optimization process. As CMA-ES is a black-box optimizer sensitive to the structure of the objective function, an ablation or sensitivity study would strengthen the work.

2. The stability and effect of the EMA mechanism deserve more in-depth analysis, particularly regarding its sensitivity to hyperparameters and its role in generalization.

3. I am not an expert in this specific subfield, so I cannot confidently assess whether similar models or ideas have previously been proposed. Additionally, while the reported results are promising, it is difficult to judge the practical significance of the improvements without deeper domain knowledge or context.

---

> ### Author Rebuttal · Authors · 2025-07-30
>
> # Response to Reviewer
>
> We thank the reviewer for the detailed feedback. We greatly appreciate your recognition of our work as offering a "compelling, efficient, and accurate" Test‑Time Adaptation framework tailored to real‑world acoustic shifts. Below, we address your questions. We will include these results and analysis in the appendix of the revised version.
>
> ---
>
> - **W1:** The proposed multi-scale loss combines utterance-level and token-level objectives, but it remains unclear how the relative weighting of these loss components affects the CMA-ES optimization process. As CMA-ES is a black-box optimizer sensitive to the structure of the objective function, an ablation or sensitivity study would strengthen the work.
>
>   **Response:** To address this, we conducted a sensitivity analysis of the loss weights α and β for the entropy loss and utterance‑level loss, which in turn leads to changes in the token‑level weight c (Minmax normalization Equation in Section 3.3). The WER results using the caf‑real condition of CHiME3 dataset are shown in Table 1 below. The results reveal that WER is remarkably stable across a wide range of weight settings. WER stays within a narrow range across different values of α and β, showing a broad area of near-optimal performance instead of a single best point. The results suggest that both components of the loss must be balanced, as extreme values for either can hurt performance. The chosen setting (α=1.0, β=2.0) sits comfortably in this stable region and consistently yields the best performance. We will include this in the appendix.
>
>
>   **Table 1.** WER (%) of Wav2Vec2ForCTC on caf‑real CHiME‑3 under varying α (rows) and β (columns), representing varying weights for the token-level loss.
>
>   | α \ β | 0.5  | 1.0  | 1.5  | 2.0  | 2.5  | 3.0  | 3.5  | 4.0  | 4.5  | 5.0  |
>   |-------|------|------|------|------|------|------|------|------|------|------|
>   | **0.5** | 37.7 | 38.2 | 38.5 | 38.6 | 39.1 | 39.4 | 39.2 | 39.8 | 39.4 | 39.7 |
>   | **1.0** | 38.1 | 38.1 | 38.0 | 37.9 | 38.4 | 38.3 | 38.4 | 38.8 | 38.7 | 39.6 |
>   | **1.5** | 38.5 | 38.1 | 38.4 | 38.1 | 38.5 | 38.0 | 38.2 | 37.8 | 38.3 | 38.4 |
>   | **2.0** | 38.4 | 38.3 | 38.0 | 37.9 | 37.9 | 38.1 | 38.0 | 38.3 | 38.0 | 38.4 |
>
> ---
>
> - **W2:** The stability and effect of the EMA mechanism deserve more in-depth analysis, particularly regarding its sensitivity to hyperparameters and its role in generalization.
>
>   **Response:** Table 2 below presents our ablation study on the EMA parameter γ, where we evaluated performance across γ ∈ {0.50, 0.60, 0.70, 0.80, 0.90, 0.95, 0.99} under the caf-real condition. We observed a clear U-shaped WER curve: performance improves up to γ = 0.90 (best WER 37.9%), and then degrades as γ increases further. This behavior aligns with our expectation and is also observed in other datasets that smaller γ values lead to unstable, overly reactive updates, while larger values result in over-smoothing and hinder adaptation. We will include these sensitivity analyses and tables in the appendix of the revised submission.
>
>   **Table 2.** WER (%) of Wav2Vec2ForCTC on caf‑real CHiME‑3 for different T‑EMA decay γ.
>
>   | γ    | 0.50 | 0.60 | 0.70 | 0.80 | 0.90 | 0.95 | 0.99 |
>   |------|------|------|------|------|------|------|------|
>   | **WER (%)** | 39.1 | 39.1 | 38.4 | 38.1 | 37.9 | 38.3 | 39.9 |
>
> ---
>
> Thank you for your comments! Let us know if we can further clarify anything.

---

> > ### Comment · Reviewer_Wcum · 2025-08-06
> >
> > Thanks for additional experiments. I'm already in positive score.

---

### Official Review · Reviewer_8uQQ · 2025-07-14

**Clarity:** 4
**Significance:** 3
**Originality:** 3
**Rating:** 5
**Confidence:** 3

**Summary:**

In this paper, the authors propose E-BATS, a novel backpropagation-free test time adaptation framework for speech fundamental models. The method utilizes a lightweight cue adaptation module that combines multi-scale distributional alignment loss (entropy, part-of-speech level, and token level) and test time-exponential moving average to achieve stable, memory-efficient adaptation under dynamic acoustic conditions. Extensive experiments on multiple benchmarks show strong performance across a wide range of domain shifts, significantly outperforming existing BP-free methods and comparable to BP-based methods, while maintaining low memory usage.

**Questions:**

1. How sensitive is the adaptation performance to the number of CMA-ES candidates and iterations? Could performance degrade with fewer samples or iterations under tighter latency budgets?
2. Is there any target domain condition where the pre-collected source stats perform poorly or even misguide the adaptation (*e.g.*, heavily accented speech)?
3. Would a learnable or data-driven initialization of prompt vectors improve convergence or stability over the mean-zero initialization used in CMA-ES?

**Ethical Concerns:**

["NO or VERY MINOR ethics concerns only"]

**Final Justification:**

My concerns have been addressed. I will keep my positive rating.

**Limitations:**

As mentioned by the authors, the CMA-ES optimization introduces latency, and GPU parallelism is not fully leveraged. This may hinder adoption in strict real-time settings.

**Paper Formatting Concerns:**

The formatting is clear and easy to read.

**Quality:**

4

**Strengths And Weaknesses:**

**Strengths**
- Clear and well-written paper with intuitive motivation and clear organization. Methodology is described in sufficient detail.
- The proposed approach addresses a practical and under-explored setting: test-time adaptation for SFMs without backpropagation.
- Significant memory savings (2x–6.4x) over BP-based methods while achieving competitive or superior performance.
- Ablation study is thorough and validates the necessity of each component (prompt adaptation, multi-scale loss, T-EMA).

**Weaknesses**
- The effectiveness of the prompt candidate search via CMA-ES may be sensitive to the complexity of the target domain. There is no discussion on candidate sufficiency, especially for domains with high variability.
- While the authors argue for the reliability of pre-collected source statistics, the domain generality or robustness of these statistics (*e.g.*, under accent or unseen noise types) is not thoroughly analyzed.
- Entropy-only optimization is known to lead to trivial solutions, and although this is addressed via auxiliary loss terms, it would help to see more quantitative evidence on how frequently trivial predictions (*e.g.*, blanks) occur and how the loss balances affect that.

---

> ### Author Rebuttal · Authors · 2025-07-31
>
> ## Response to Reviewer
> We thank the reviewer for the very detailed feedback. We greatly appreciate the reviewer recognition of our work addressing "a practical and under-explored setting" of the backpropagation-free TTA framework for speech foundation model with "competitive or superior performance" and "significant memory saving." Below, we address the reviewer's concerns, and will include these results and analysis in the appendix of the revised version.
>
> ---
>
> - **W1:** The effectiveness of the prompt candidate search via CMA-ES may be sensitive to the complexity of the target domain. There is no discussion on candidate sufficiency, especially for domains with high variability.
>
>   **Response:** We selected four target domains exhibiting increasing variability, quantified by the covariance shift offered by Fréchet Inception Distance (FID) of the embedding distribution between source and target. These range from low to high variability: Gaussian noise (σ = 0.01), CHiME-3 café-simulated, CHiME-3 café-real, and CHiME-3 dynamic. As shown in Table 1 below, with CMA-ES population size increasing from 10 to 100, the largest gains occur moving from 10 to 40 candidates across all domains. Beyond 50 candidates, performance plateaus. These results show that while more complex domains (café-real, dynamic) begin at higher absolute WER, they exhibit the same early-saturation behavior as simpler conditions. Therefore, sensitivity to domain complexity may not be directly related to the number of prompt candidates considered. We will include this ablation in the appendix.
>
>   **Table 1.** WER (%) across different candidate sizes on four target domains with varying complexity.
>
>   | Candidate Size | 10   | 20   | 30   | 40   | 50   | 60   | 70   | 80   | 90   | 100  |
>   |---------------|------|------|------|------|------|------|------|------|------|------|
>   | **Gaussian (σ=0.01)**                | 16.2 | 15.6 | 15.2 | 14.9 | 14.8 | 14.8 | 14.7 | 14.7 | 14.7 | 14.7    |
>   | **CHiME-3 Single (café-simulated)**  | 17.0 | 16.5 | 16.5 | 16.2 | 16.1 | 16.3 | 16.1 | 16.0 | 16.0 | 16.2 |
>   | **CHiME-3 Single (café-real)**       | 41.8 | 39.3 | 38.3 | 38.3 | 37.9 | 38.0 | 37.9 | 37.9 | 38.0 | 37.8 |
>   | **CHiME-3 Dynamic**                  | 25.7 | 25.1 | 24.7 | 24.6 | 24.3 | 24.3 | 24.4 | 24.2 | 24.4 | 24.3 |
>
> ---
>
> - **W2:** While the authors argue for the reliability of pre-collected source statistics, the domain generality or robustness of these statistics (e.g., under accent or unseen noise types) is not thoroughly analyzed.
>
>   **Response:** Our pre-collected source statistics are derived from the clean LibriSpeech dataset, which serves as the training data for SFMs. In our experiments, they consistently improved adaptation without overfitting, even under diverse accents and noise types. We would appreciate any additional clarification on this weakness.
>
> ---
>
> - **W3:** Entropy-only optimization is known to lead to trivial solutions, and although this is addressed via auxiliary loss terms, it would help to see more quantitative evidence on how frequently trivial predictions (e.g., blanks) occur and how the loss balances affect that.
>
>   **Response:** We analyzed and compared the occurrence of trivial predictions under two conditions: (i) entropy-only and (ii) entropy with utterance-level loss, using the CHiME-3-Mix dataset. As shown in Table 2, we observed two types of trivial solutions: blank predictions and single-character predictions (entropy = 0 with only one character predicted). It is evident that the utterance-level loss significantly reduced the incidence of trivial solutions from 37.5% to 0.61%, thereby improving the WER. We will include this analysis in the appendix.
>
>   **Table 2.** Trivial solutions on CHiME-3-Mix: number and percentage of trivial predictions among all predictions.
>
>   | **Configuration**           | **Blank**           | **Single-char**         | **Total problematic**     | **WER (%)** |
>   |-----------------------------|---------------------|--------------------------|----------------------------|-------------|
>   | **Entropy-only**            | 42 (1.59%) &emsp;   | 948 (35.91%) &emsp;     | 990 (37.50%) &emsp;       | 49.6        |
>   | **+ Utterance-level loss**  | 12 (0.45%) &emsp;   | 4 (0.15%) &emsp;        | 16 (0.61%) &emsp;         | 25.5        |
>
>
> ---
>
> - **Q1:** How sensitive is the adaptation performance to the number of CMA-ES candidates and iterations? Could performance degrade with fewer samples or iterations under tighter latency budgets?
>
>   **Response:** We find the expected trade-off between the candidate numbers and optimization iterations with word error rate (WER) (Table 3 and Table 4 below). We will add a sentence in the main text and include these results in the appendix. As shown in Table 3 (caf-real condition of CHiME-3 using Wav2Vec2ForCTC), increasing the CMA-ES candidate size from 10 to 40 yields the largest gains. Table 4 demonstrates that moving from 5 to 25 iterations captures most of the benefit. We observe the same early-saturation behavior across other conditions (e.g., bus-real, street-real) and datasets.
>
>   **Table 3.** WER (%) of Wav2Vec2ForCTC on the _caf-real_ condition of CHiME-3 under different CMA-ES candidate sizes (iteration steps = 25).
>
>   | Candidate Size | 10   | 20   | 30   | 40   | 50   | 60   | 70   | 80   | 90   | 100  |
>   |--------------------|------|------|------|------|------|------|------|------|------|------|
>   | **WER (%)**        | 41.8 | 39.3 | 38.3 | 38.3 | 37.9 | 38.0 | 37.9 | 37.9 | 38.0 | 37.8 |
>
>   **Table 4.** WER (%) of Wav2Vec2ForCTC on the _caf-real_ condition of CHiME-3 under different CMA-ES iteration steps (candidate size = 50).
>
>   | Iteration Steps | 5    | 10   | 15   | 20   | 25   | 30   | 35   | 40   | 45   | 50   |
>   |-----------------|------|------|------|------|------|------|------|------|------|------|
>   | **WER (%)**     | 40.8 | 39.9 | 38.6 | 38.0 | 37.9 | 38.0 | 37.9 | 37.6 | 38.4 | 38.0 |
>
> - **Q2:** Is there any target domain condition where the pre-collected source stats perform poorly or even misguide the adaptation (e.g., heavily accented speech)?
>
>   **Response:** We clarify that the pre-collected statistics are derived from clean speech rather than noisy speech, and therefore are not misleading in any adaptation scenario. Specifically, SFMs are trained on clean speech using publicly available datasets such as LibriSpeech, and the pre-collected statistics are obtained from the clean LibriSpeech dataset. As a result, these statistics do not introduce noise or negatively impact the adaptation process. In the revised version, we will include a sentence explicitly stating that the pre-collected statistics are sourced from clean LibriSpeech data.
>
> - **Q3:** Would a learnable or data-driven initialization of prompt vectors improve convergence or stability over the mean-zero initialization used in CMA-ES?
>
>   **Response:** Thank you for the suggestion. While we currently use a mean-zero initialization in CMA-ES for simplicity and generality, we agree that learnable or data-driven initialization (e.g., from previous prompts, pre-trained embeddings, or lightweight estimators) could potentially improve convergence speed and stability, especially in early adaptation steps. This is an interesting direction that complements our T-EMA design and will be explored in future work.
>
> ---
>
> Thanks again for your comments! We remain open to addressing any additional concerns or clarifications you may have.

---

> > ### Comment · Reviewer_8uQQ · 2025-08-08
> >
> > Thank you for your rebuttal and additional experiments. My concerns have been addressed. I will keep my positive rating.

---

### Official Review · Reviewer_uVzm · 2025-07-22

**Clarity:** 3
**Significance:** 2
**Originality:** 3
**Rating:** 3
**Confidence:** 3

**Summary:**

This paper proposes a novel backpropagation-free adaptation approach specifically designed for CTC (Connectionist Temporal Classification) models that incorporate pre-trained speech encoders. The method addresses a critical challenge in speech recognition: efficiently adapting models to new acoustic conditions and domains without the computational overhead of traditional gradient-based fine-tuning. The paper further introduces a sampling strategy over a set of learnable parameters, guided by a combination of losses at different levels. Experimental results are promising in terms of both accuracy and GPU memory efficiency.

**Questions:**

[1] Misuse of “blank” tokens:
- The paper repeatedly discusses “blank” tokens in the context of speech foundation models. However, speech foundation models typically refer to encoders pre-trained via self-supervised learning (e.g., wav2vec, HuBERT), which do not use blank tokens. Blank tokens are specific to CTC or Transducer objectives. While the authors seem to apply a CTC objective during fine-tuning or adaptation, this is never clearly stated in the method section. CTC is only vaguely mentioned in the experimental setup, making the discussion around blanks confusing and misleading.

[2] Misleading scope and title:
-  The title and scope suggest a method tailored to “speech foundation models,” but the proposed method is in fact specific to models trained with the CTC objective. The foundation model merely serves as the encoder. CTC models and speech foundation models are conceptually distinct, and the proposed technique is not unique to foundation models—it could be applied to any encoder trained with CTC. Therefore, the title should reflect the method’s actual scope, i.e., adaptation for CTC models, not speech foundation models.

[3] Incorrect explanation of blank tokens:
- The statement that blank tokens represent “frames where no alphabetic character can be assigned, typically due to ambiguous sounds or silent periods” is inaccurate. In CTC, blank tokens address the alignment mismatch between input and output lengths and are essential to allowing flexible alignments—not limited to silences or ambiguities.

[4] Prompt tuning claims:
-  The authors write, “While conventional prompt tuning approaches have primarily been developed for transformer-only architectures…,” which is misleading. In speech processing, parameter-efficient adaptation using small learnable modules predates the rise of transformers. Concepts like adapter layers or bottleneck modules were already in use with LSTM or CNN-based models. Therefore, the claim about prompt tuning being “primarily developed for transformers” is historically inaccurate.

**Ethical Concerns:**

["NO or VERY MINOR ethics concerns only"]

**Final Justification:**

I thank the authors for their great attempt in addressing the concerns, however, reading all reviews and comments I will keep my initial score.

**Limitations:**

[1] On mean shift:
-  The paper attributes distribution shifts mainly to mean shifts. However, most of the noise scenarios appear to be simple additive noise, where mean shift is expected. To strengthen this claim, the authors should test their method on other types of distortions (e.g., reverberation, channel effects) to confirm whether mean shift still dominates.

[2] On token-level loss and pseudo-labels:
- The token-level loss relies on pseudo-labels, which may be incorrect. If a wrong label has a source-domain-like hidden representation, the adaptation may reinforce the wrong prediction. This risk is acknowledged but not addressed convincingly. Moreover, the ablation suggests the token-level loss is not particularly effective. This should be discussed more critically.

[3] Entropy loss risk:
-  The same risk applies to the entropy loss. Encouraging low entropy for incorrect predictions may further entrench errors. More analysis is needed to understand the failure modes of this loss.

[4] Missing ablation on utterance-level loss:
-  The utterance-level loss appears to be the most reasonable component, aligning overall acoustic features like speaker or noise condition. However, the paper does not present an ablation study on using only this loss, which would help clarify its actual contribution.

[5] Choice of LibriSpeech subsets:
- The authors use LibriSpeech test-other, which is already relatively noisy. Why not use test-clean and apply synthetic noise to demonstrate the effectiveness more clearly?

[6] CTC model details missing:
It is unclear how the CTC models are obtained. Were they fine-tuned from foundation models by the authors? Or are they taken from public checkpoints (e.g., Hugging Face, Fairseq)? This detail is essential for reproducibility and understanding the setup.

[7] CommonVoice version:
- The version of CommonVoice used in the experiments is not reported. This should be clearly specified.

[8] Adaptation speed:
-  There is no measurement of adaptation time. Since the paper positions itself as a test-time method, it would be helpful to include comparisons of time efficiency against baseline approaches

**Quality:**

3

**Strengths And Weaknesses:**

Strengths:
- The use of sequence-level loss is well-motivated.
- Low-level acoustic factors such as background noise, speaker characteristics, and environmental conditions typically affect entire speech sequences uniformly rather than individual tokens is sound. It makes sequence-level alignment and adaptation a reasonable and effective methodological choice that aligns with the underlying nature of acoustic domain shifts.

Weaknesses:
- There are several inaccuracies and conceptual issues in the method section regarding CTC, speech foundation models, and adaptation.

---

> ### Author Rebuttal · Authors · 2025-07-31
>
> ## Response to Reviewer
> We thank the reviewer for the very detailed feedback. We greatly appreciate the reviewer acknowledges the novelty and motivation of our backpropagation-free TTA method, highlighting its effectiveness in handling domain shifts through sequence-level loss and its efficiency in both accuracy and memory usage. Below, we address the reviewer's concerns, and we will include these results and analysis in the appendix of the revised version.
>
> ---
> - **Q1:** Misuse of blank tokens
>
>   **Response:** Blank tokens are standard in CTC-based ASR models, and CTC objective is widely used in ASR systems. We will move the CTC setup earlier in Section 3.1, before detailing the method to clarify the scope. Importantly, the novelty of our proposed TTA method remains intact: our approach is specifically designed for speech foundation model encoder architectures, and to our knowledge, has not been addressed in previous work. We will revise Section 3.1 to clarify the scope with respect to CTC objectives.
>
> ---
> - **Q2:** Misleading scope and title
>
>   **Response:** Thanks for bringing the discussion of the scope. Our method is fundamentally driven by the encoder architecture of speech foundation models (SFMs), but not the CTC objectives, therefore, we would keep the title and scope unchanged. Every component of E‑BATS, from injecting learnable prompts into the final CNN feature maps to applying our multi‑scale discrepancy loss across each transformer block, directly leverages the SFM encoder design. To further test the generality of our method beyond ASR with CTC, we applied E‑BATS to speech emotion recognition tasks (cross-entropy loss) using the IEMOCAP dataset and the `SpeechBrain/emotion-recognition-wav2vec2-IEMOCAP` model under additive Gaussian noise (σ=0.02). Adaptation increased emotion prediction accuracy from 38.8% to 43.4%, demonstrating that our E-BATS framework can enhance the performance of other downstream tasks using SFMs. We will include the results in the appendix.
> ---
> - **Q3** Incorrect explanation of blank tokens
>
>   **Response:** We appreciate the chance to clarify. We acknowledge that blank tokens address the alignment mismatch in CTC tasks. However, we have observed the predictions in both silent and ambiguous sounds empirically in our experiments and thus mentioned them for simplicity. Moreover, our observations align with recent studies that have claimed that predicting blank tokens in CTC helps eliminate the need to identify inherently ambiguous label boundaries [1], and also highlights silent regions [2]. We will revise the text to include the claim regarding alignment to address this concern.
>
> ---
> - **Q4:** Prompt tuning claims
>
>   **Response:** We'd like to highlight "prompt tuning" and "adapter layers or bottleneck modules" are two distinct parameter-efficient fine-tuning approaches [3]. Prompt tuning, introduced by Lester et al. [4], adds a small set of learnable “soft” tokens to the input sequence, which are learned via the Transformer’s attention mechanism. This allows improved performance by updating those prompt embeddings directly without modifying the model’s original weights. In contrast, adapter layers insert bottleneck modules within the network and train their internal weights and biases. In our work, we refer specifically to the modern concept of prompt tuning, which is primarily designed for Transformers. To address the reviewer’s concern, we will include references [3, 4] in Section 2 for clarity.
>
> ---
> - **W1:** On mean shift.
>
>   **Response:** CHIME-3, TEDLIUM, and Common Voice datasets contains real-world environmental noise. Specifically, the CHIME-3 dataset includes recordings from four distinct acoustic environments: bus, café, street junction, and pedestrian area. Each of these presents unique natural reverberation characteristics, from the reflective interiors of buses and cafés to the more open acoustics found in street junctions and pedestrian areas. While including a broader range of conditions would be ideal, we believe that the 16 conditions investigated are sufficient to draw a conclusive result. We plan to address this in future work as more real-world datasets with reverberation become available, and appreciate suggestions from the reviewer regarding the dataset.
>
> ---
> - **W2:** On token-level loss and pseudo-labels.
>
>   **Response:** We address this with the proposed adaptive weighting mechanism for the token-level loss, as described in Section 3.3. Specifically, the weight assigned to the token-level loss depends on the reliability of the pseudo-labels, which is measured by the entropy and the utterance-level loss. If the sum of the entropy and utterance-level loss is low, indicating more reliable pseudo-labels, we assign a higher weight to the token-level loss, and vice versa. This approach is intended to mitigate the risk of relying on incorrect pseudo-labels.
>
> Regarding effectiveness, in our design, token-level loss is incorporated to provide valuable support when distribution shifts are relatively minor or when pseudo labels are more reliable. We recognize that in severe scenarios requiring substantial adaptation, utterance-level loss and entropy serve as the primary and most effective drivers of adaptation. The proposed multi-level loss aims to address both subtle and significant distribution shifts.
>
> ---
> - **W3:** Entropy loss risk.
>
>   **Response:** We analyzed the failure mode of entropy loss and further demonstrated that the proposed utterance-level loss can compensate for it. The results are presented in Table 1, using CHiME-3-Mix dataset. Entropy loss leads to blank token predictions and single-character predictions (entropy = 0 with only one character predicted), accounting for 37.5% of cases. By incorporating utterance-level loss, these problematic predictions are significantly reduced, resulting in a substantial improvement in WER. We will include a detailed analysis in the appendix.
>
>   **Table 1.** Trivial solutions on CHiME-3-Mix: number and percentage of trivial predictions among all predictions.
>   |Configuration|Blank|Single-char|Total|WER (%)|
>   |-|-|-|-|-|
>   |Entropy-only|42 (1.59%)&emsp;|948 (35.91%)&emsp;|990 (37.50%)&emsp;|49.6|
>   |+Utterance-level loss|12 (0.45%)&emsp;|4 (0.15%)&emsp;|16 (0.61%)&emsp;|25.5|
>
> ---
> - **W4:** Missing ablation on utterance-level loss.
>
>   **Response:** We evaluated performance using only the utterance-level loss under the CHiME3-mix condition with Wav2Vec2ForCTC and achieved a WER of 26.8%. This result surpasses the performance with entropy loss alone and we appreciate the reviewer’s suggestion. We will include this additional ablation study in Table 3. Nonetheless, combining both losses yields even greater improvements of 25.5%.
>
> ---
> - **W5:** Choice of LibriSpeech subsets.
>
>   **Response:** To enable a fair and direct comparison with baselines like SUTA and CEA, we used the same LibriSpeech test-other set for evaluation.
> ---
> - **W6:** CTC model details missing.
>
>   **Response:** All CTC models used in our experiments are based on publicly available checkpoints from Facebook via Hugging Face, specifically `facebook/wav2vec2-base-960h` and `facebook/hubert-large-ls960-ft`. Moreover, as noted at the end of the abstract, we will release our code and full reproduction instructions upon acceptance to ensure complete transparency and reproducibility.
>
> ---
> - **W7:** CommonVoice version.
>
>   **Response:** As presented in Appendix B.1, we adopted the test set from the en-June-22nd-2020 version which will be included as footnote in the revised version.
>
> ---
> - **W8:** Adaptation speed
>
>   **Response:** We compute the per-utterance time complexity to compare E‑BATS with the top-performing backpropagation-based (DSUTA) and backpropagation-free (FOA) methods (see Table 2). DSUTA scales with the large number of model parameters via backpropagation (P), often in hundreds of millions, whereas E‑BATS only involves 𝒪(d²) and 𝒪(d³) operations (with d=512). Since P ≥ d³ in most SFM models, E‑BATS achieves significant efficiency gains over DSUTA. While both FOA and E-BATS share the same asymptotic complexity of 𝒪(d³) for CMA-ES updates, E-BATS are practically more efficient since FOA needs a 3³ multiplicative factor of 𝒪(d³) as it requires three prompts for adaptation. And FOA incurs an additional 𝒪(Ld²) cost (over d) per prompt for prompt attention with L transformer layer encoders. We will include the analysis in the appendix. Nonetheless, our main focus, as noted in the introduction, is the trade-off between accuracy and memory efficiency, which is more critical for resource-constrained devices.
>
>   **Table 2**: Per-utterance adaptation time complexity (big-O) for E-BATS and top TTA baselines.
>   |Method|Time Complexity per Utterance|
>   |-|-|
>   |DSUTA|𝒪(N × (F + E + P))|
>   |FOA|𝒪(N × [J (F + E + L d²) + d³])|
>   |E-BATS|𝒪(N × [J (F + E + L d + L\|V\|d) + d³])|
>
>   where:
>   - F: cost of one forward pass
>   - E: cost of entropy loss calculation
>   - N: number of adaptation iteration steps per utterance
>   - J: number of candidate prompts per iteration
>   - |V|: size of the token class
>
> ---
> References:
>
> [1] Graves, A. ,et al . Connectionist Temporal Classification: Labelling Unsegmented Sequence Data with Recurrent Neural Networks. ICML 2006.
>
> [2] Mostafa, A., et al. Phonological Level Wav2Vec2‑based Mispronunciation Detection and Diagnosis Method. *Journal of Speech Technology*, 10(2), 123–135.
>
> [3] Wang, L., et al. Parameter-efficient fine-tuning in large language models: a survey of methodologies. Artif Intell Rev 58, 227 (2025).
>
> [4] Lester, B.,et al. The Power of Scale for Parameter‐Efficient Prompt Tuning. In EMNLP 2021.

---

> > ### Author Response · Authors · 2025-08-07
> >
> > Dear reviewer. Thank you for reading our rebuttal! We believe that our response addresses your raised weaknesses and questions. If you have any outstanding concerns, please let us know so that we can do our best to address them.

---

### Note · Authors · 2025-08-13

We thank the reviewers and the Area Chair for their time and thoughtful feedback, particularly their recognition of our work as:
* “the first to propose a backpropagation-free TTA method specifically for speech foundation models” (LXCm)
* “provides a compelling, efficient, and accurate alternative for speech scenarios” (Wcum)
* “achieving strong accuracy… while using significantly less memory” (4Yrj)
* “addresses a practical and under-explored setting: test-time adaptation for SFMs without backpropagation” (8uQQ)

During the rebuttal phase, we carefully addressed all points raised by the reviewers and will include these clarifications in the revised version and appendix.
* **Scope and CTC Loss** (uVzm): We clarified that our method is not restricted to CTC-based approaches. Our method is designed for speech foundation models (see Section 3) and are not constrained to CTC loss. We further tested our method on a different task, speech emotion recognition (SER), using cross-entropy loss, which confirms our method works beyond ASR and CTC.
* **Generalisability** (4Yrj, LXCm): We carried out extra experiments on non-ASR tasks (SER), showing that our method is flexible and effective in other settings.
* **Detailed Ablation Studies and Loss Function Analyses** (8uQQ, Wcum): We have included further detailed ablation studies on hyperparameters, expanding upon Table 3 and Section 5.5, which provide additional insights and justification for our hyperparameter choices.
* **Clarification of Misunderstandings** (uVzm): We addressed misunderstandings regarding the differences between prompt tuning and adapter layers, as well as the handling of blank and special tokens, and will include additional references in the revised version.

While we appreciate that reviewers acknowledged our responses and did not raise further questions, we unfortunately did not have the opportunity to engage further with reviewer uVzm. Nevertheless, we are confident that our clarifications and expanded analyses thoroughly address the points raised in the reviews.

---

### Decision · Program_Chairs · 2025-09-17

**Decision:**

Accept (poster)

**Comment:**

This paper addresses the problem of backpropagation-free test-time adaptation, which the AC considers a highly practical and necessary direction for broad real-world applications of TTA. The proposed approach builds on the solid foundation of FOA and extends it to speech foundation models.

The overall novelty and technical contributions are acknowledged by four of the five reviewers, as well as the AC. Although Reviewer uVzm raised some concerns, he/she did not provide detailed follow-up feedback after the rebuttal. The AC carefully reviewed the paper and examined the author's response to the raised issues by Reviewer uVzm. The AC finds that most concerns have been addressed and therefore recommends acceptance of this paper.

The authors are kindly reminded to incorporate the new rebuttal results and discussions into the final version.